# Exploring Neural Network Landscapes: Star-Shaped and Geodesic Connectivity

## Abstract

One of the most intriguing findings in the structure of neural network landscapes is the phenomenon of *mode connectivity* Freeman & Bruna (2017); Draxler et al. (2018): For two typical global minima, there exists a path connecting them without barrier. This concept of mode connectivity has played a crucial role in understanding important phenomena in deep learning.

In this paper, we conduct a fine-grained analysis of this connectivity phenomenon. First, we demonstrate that in the overparameterized case, the connecting path can be as simple as a *two-piece linear path*, and the path length can be nearly equal to the Euclidean distance. This finding suggests that the landscape should be nearly convex in a certain sense. Second, we uncover a surprising *star-shaped* connectivity: For a finite number of typical minima, there exists a center on the minima manifold that connects all of them simultaneously via linear paths. These results are provably valid for linear networks and two-layer ReLU networks under a teacher-student setup, and are empirically supported by models trained on MNIST and CIFAR-10.

## 1 Introduction

It is well-known that neural networks are highly non-convex, but they can still be efficiently trained by simple algorithms like stochastic gradient descent (SGD). Understanding the underlying mechanism is crucial and in particular, a key aspect of this is to uncover the topology and geometry of neural network landscapes.

Some recent studies exploited the local properties of neural network landscapes, including the absence of spurious minima Ge et al. (2016); Soudry & Carmon (2016), the sharpness of different minima Hochreiter & Schmidhuber (1997); Keskar et al. (2017); Wu et al. (2017), and the structures of saddle points Zhang et al. (2021) and plateaus Ainsworth & Shin (2021). Other studies have examined the nonlocal structures, including the impact of symmetries and invariances Simsek et al. (2021), the presence of non-attracting regions of minima Petzka & Sminchisescu (2021), the monotonic linear interpolation phenomenon Goodfellow et al. (2014); Wang et al. (2022); Vlaar & Frankle (2022); Lucas et al. (2021). Among these nonlocal structures, one of the most intriguing findings is the *mode connectivity*, which is the focus of this paper.

Mode connectivity refers to the property that global minima of (over-parameterized) neural networks are (nearly) path-connected and form a connected manifold Cooper (2018), rather than being isolated. This characteristic of neural network landscape was first observed by Freeman and Bruna in Freeman & Bruna (2017), and its practical universality was later demonstrated in Draxler et al. (2018); Garipov et al. (2018) through extensive large-scale experiments. Mode connectivity has attracted wide attention and has been utilized to understand many important aspects of deep learning, including the role of permutation invariance Entezari et al. (2021); Ainsworth et al. (2022), properties of SGD solutions Mirzadeh et al. (2020); Frankle et al. (2020), explaining the success of certain learning methods such as ensemble methods Garipov et al. (2018); He et al. (2019), and even to design methods with better generalization Benton et al. (2021).

In this paper, we conduct a fine-grained analysis of this mode connectivity. Our specific investigation is inspired by the empirical observation that *the connecting paths can be piecewise linear with as few as two*

*pieces* Garipov et al. (2018). This motivates us to examine the piecewise linear connectivity of the global minima manifold. Two minima are said to be *k*-piece linearly connected if they can be connected using paths with at most *k* linear segments. Specifically, our main contributions are summarized as follows.

- We provide a theoretical analysis of the piecewise linear connectivity for two-layer ReLU networks and linear networks under a teacher-student setup. We prove that as long as the network is sufficiently over-parameterized, any two minima are 2-piece linearly connected. By exploiting this property, we further discover the following surprising structures of the global minima manifold:

    - **Star-shaped connectivity:** For a finite set of typical minima, there exists a minimum (center) such that it is linearly connected to all these minima simultaneously.
    - **Geodesic connectivity:** For two typical minima, the geodesic distance on the minima manifold is close to the Euclidean distance. Moreover, the ratio between them monotonically decreases towards 1 as increasing the network width. This suggests that the landscape of over-parameterized networks might be not far away from a convex one in some sense.

- We then provide extensive experiments on MNIST and CIFAR-10 datasets that confirm our theoretical findings on practical models.

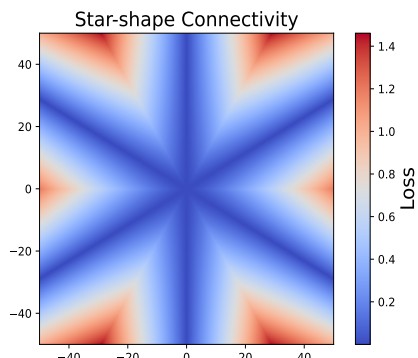
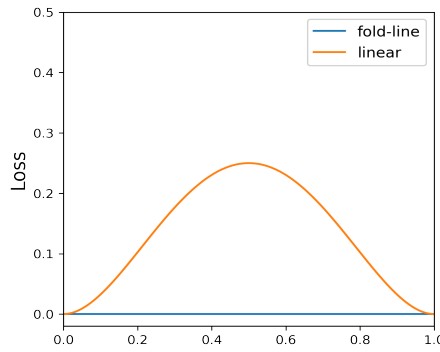

**Figure 1: Left**: The speculation of a potential shape of the star-shaped connectivity in the loss landscape. Due to the limitation in 2-dimensional visualization, here we only provide a potential section as a heuristic plot. **Right**: For 2 minima $\theta_1, \theta_2$ as described in the setting of Proposition 16, we consider the linear mode connectivity through a center $\theta^*$. For the linear interpolations between two minima, and the $\theta_1 \to \theta^* \to \theta_2$ fold-lines constructed by two linear interpolations, we plot the training loss along these paths. Specifically, the $x$-axis $t$ here denotes the point $t\theta^* + (1-t)\theta_i$ in the linear interpolation (the orange line). On the other hand, for the loss along the fold-line (blue line), $t < 0.5$ corresponds to the point $2t\theta^* + (1-2t)\theta_1$, while $t \geq 0.5$ corresponds to $(2t-1)\theta_2 + (2-2t)\theta^*$. The result shows our expectation of linear mode connectivity through the center we obtained.

## 1.1 Related works

**Understanding the mode connectivity.** There have been several studies that aim to theoretically explain the phenomenon of mode connectivity. The initial work by Freeman and Bruna Freeman & Bruna (2017) proved the mode connectivity for both linear networks and two-layer ReLU networks when the model is regularized by squared $\ell_2$ norm. Garipov et al. (2018) empirically discovered that connecting paths can be piecewise linear. Based on this observation, Kuditipudi et al. (2019) proved that if two minima satisfy certain conditions such as the dropout stability and noise stability, then they can be connected using paths with at most 10 linear segments. Moreover, the result in Kuditipudi et al. (2019) is applicable to deep neural networks. Shevchenko & Mondelli (2020) provided a theoretical explanation of why SGD tends to find solutions that satisfy the dropout stability for two-layer neural networks. In addition to these studies, Nguyen et al. (2018; 2021a;b) investigated how the connectivity depends on the network width and depth, but the analysis does not provide any information about the structure of the connecting paths. In contrast, we

investigate the particular structure of connecting paths and provide both empirical and theoretical evidence showing that when networks are sufficiently over-parameterized, typical minima can be connected using paths of merely 2 linear segment. Moreover, we also explore the geometry of the global minima manifold by using the simplicity of connecting paths.

**Critical points.** Fukumizu et al. (2019); Zhang et al. (2021) studied the hierarchical structures of critical points and in particular, how local minima degenerate to saddle points when increasing the number of neurons. Ros et al. (2019); Maillard et al. (2020) provided an analytical characterization of the distribution of critical points for learning a single neuron in an asymptotic regime by using the Kac-Rice replicated method from statistical physics. Ainsworth & Shin (2021); Fukumizu & Amari (2000); Yoshida & Okada (2019) studied the appearance of plateaus in the neural network landscape and its impact on the training process. In addition, it has been always a major problem to understand under what conditions bad local minima/valley disappear Kawaguchi (2016); Soudry & Carmon (2016); Liang et al. (2018); Lin et al. (2022) or not Auer et al. (1995); Safran & Shamir (2018); Yun et al. (2018); Ding et al. (2022); Lu & Kawaguchi (2017). Another line of works inspects curvatures at minima Sagun et al. (2017) and how the curvatures of the local landscape are related to the generalization of networks represented by those minima Hochreiter & Schmidhuber (1997); Wu et al. (2017); Jastrzębski et al. (2017); Ma & Ying (2021); Wu et al.. In this paper, we study the topology and geometry of global minima by utilizing the mode connectivity.

**Nonlocal structures.** Note that the mode connectivity is nonlocal in nature. Therefore, our work is also helpful for understanding the nonlocal structures of the neural network landscape. Fort & Jastrzebski (2019) proposed a phenomenological model (a set of high dimensional wedges) to study large-scale structures of neural network landscape. Goodfellow et al. (2014) discovered the surprising monotonic linear interpolation (MLI) phenomenon: the loss often decreases monotonically in the linear interpolation between random initialization and minima found by SGD. Wang et al. (2022); Vlaar & Frankle (2022); Lucas et al. (2021) provided theoretical analyses of MLI phenomenon. Cooper (2018) proved that in the over-parameterized case, global minima form a high-dimensional manifold and this minima manifold is path connected as implied by the mode connectivity phenomenon Garipov et al. (2018); Freeman & Bruna (2017); Draxler et al. (2018). Recently, Annesi et al. (2023) showed a star-shaped structure in the regime of the spherical negative perceptron. We instead provide both theoretical and empirical evidence, showing that star-shaped structure also exists for neural networks.

## 2 Preliminaries

**Notation.** For an integer $n$, let $[n] = \{1, 2, \ldots, n\}$. For a compact $\Omega$, denote by $\mathrm{Unif}(\Omega)$ the uniform distribution over $\Omega$. Let $\mathbb{S}^{d-1} = \{\mathbf{x} \in \mathbb{R}^d : \|\mathbf{x}\|_2 = 1\}$ and $\tau_{d-1} = \mathrm{Unif}(\mathbb{S}^{d-1})$. Denote by $\{\mathbf{e}_j\}_{j=1}^d$ the canonical basis of $\mathbb{R}^d$.

Let $f : \mathcal{X} \times \Theta \mapsto \mathcal{Y}$ be a neural network with $\mathcal{X}$ and $\Theta$ denoting the input space and parameter space, respectively. Let $\ell : \mathcal{Y} \times \mathcal{Y} \mapsto \mathbb{R}$ be a loss function. Then the loss landscape is determined by $\mathcal{R}(\theta) = \mathbb{E}_{(\mathbf{x},y)\sim\mathcal{D}}[\ell(f(\mathbf{x}; \theta), y)]$, where $\mathcal{D}$ denotes the input distribution. In this paper, we make the over-parameterization assumption: $\inf_{\theta \in \Theta} \mathcal{R}(\theta) = 0$. Then, the global minima manifold is given by

$$\mathcal{M} = \{\theta \in \Theta : \mathcal{R}(\theta) = 0\}. \tag{1}$$

For any $\theta_1, \theta_2 \in \mathcal{M}$, denote by $\mathcal{P}_{\theta_1, \theta_2}$ the space of paths on $\mathcal{M}$ connecting $\theta_1$ and $\theta_2$:

$$\mathcal{P}_{\theta_1, \theta_2} = \{\gamma : [0, 1] \mapsto \mathcal{M} \mid \gamma(0) = \theta_1, \gamma(1) = \theta_2\}.$$

Existing works on mode connectivity imply that under some conditions, $\mathcal{M}$ is path connected, i.e., $\mathcal{P}_{\theta_1, \theta_2} \neq \emptyset$ for typical $\theta_1, \theta_2 \in \mathcal{M}$. In this paper, we make a refined analysis of the connectivity. To be rigorous, we formalize some concepts that we shall use throughout this paper below.

**Definition 1** (Linear interpolation). For any $\theta_1, \theta_2 \in \Theta$, denote by $\gamma^{\mathrm{lin}}_{\theta_1, \theta_2} : [0, 1] \to \Theta$ the linear interpolation path defined as $\gamma(t) = t\theta_1 + (1 - t)\theta_2$, $t \in [0, 1]$.

**Definition 2** (k-piece linear connectivity). For any $\theta_1, \theta_2 \in \mathcal{M}$, we write $\theta_1 \leftrightarrow \theta_2$ if $\gamma^{\text{lin}}_{\theta_1,\theta_2} \subset \mathcal{M}$. We say $\theta_1$ and $\theta_2$ are $k$-piece linearly ($k$-PL) connected if there exist $\beta_1, \ldots, \beta_{k-1} \in \mathcal{M}$ such that $\theta_1 \leftrightarrow \beta_1 \leftrightarrow \cdots \leftrightarrow \beta_{k-1} \leftrightarrow \theta_2$. Particularly, the case of $k = 1$ is referred to as linear connectivity.

**Definition 3** (Star-shaped linear connectivity). For multiple minima $S = \{\theta_i\}_{i=1}^r \subset \mathcal{M}$, we refer to the star-shaped linear connectivity as there exists a $\theta^* \in \mathcal{M}$ such that $\theta_i \leftrightarrow \theta^*$ for all $i = 1, 2, \ldots, r$. Specifically, $\theta^*$ and $S$ are said to be the *center* and feet, respectively.

In this paper, we also consider another quantity to measure the strength of connectivity.

**Definition 4** (Normalized geodesic distance (NGD)). For any $\theta_1, \theta_2 \in \mathcal{M}$, define the normalized geodesic distance between $\theta_1$ and $\theta_2$ by

$$G(\theta_1, \theta_2) = \frac{\inf_{\gamma \in \mathcal{P}_{\theta_1,\theta_2}} \int_0^1 \|\gamma'(t)\|_2 \, \mathrm{d}t}{\|\theta_1 - \theta_2\|_2}. \tag{2}$$

If $\mathcal{P}_{\theta_1,\theta_2}$ is an empty set, set $G(\theta_1, \theta_2) = +\infty$.

If the landscape is convex, it is trivial that the NGD is exactly 1 for any pair of global minima since the geodesic is simply the linear interpolation. However, for nonconvex landscapes, the NGD is always strictly greater than 1. The value of NGD can serve as a factor to quantify the degree of non-convexity. If the NGD keeps close to 1 for any pair of minima, then the landscape should be somehow as benign as a convex one. Otherwise, the landscape must be highly non-convex. We are particularly interested in how the value of NGD changes as increasing the level of over-parameterization.

## 3 Two-layer ReLU networks

We first consider the two-layer ReLU network under a teacher-student setup, where the label is generated by a teacher network: $f^*(\mathbf{x}) = \sum_{j=1}^M \sigma(\mathbf{w}_j^* \cdot \mathbf{x})$. Here the activation function $\sigma : \mathbb{R} \mapsto \mathbb{R}$ is given by $\sigma(z) := \max(0, z)$. We make the following assumption.

**Assumption 5.** Suppose $M \leq d$, $\mathbf{w}_j^* = \mathbf{e}_j$ for $j = 1, \ldots, M$, and $\mathbf{x} \sim \tau_{d-1}$.

By the rotational symmetry, the specific assumption that $\mathbf{w}_j^* = \mathbf{e}_j$ for $j = 1, \ldots, M$ is equivalent to only assume $\{\mathbf{w}_j^*\}_{j=1}^M$ to be orthonormal. However, we will focus on Assumption 5 to make our statement more succinct. In such a case, the loss objective can be rewritten as

$$\mathcal{R}(\theta) = \mathbb{E}_{\mathbf{x} \sim \tau_{d-1}} \left[ \left( \sum_{i=1}^m \sigma(\mathbf{w}_i \cdot \mathbf{x}) - \sum_{i=1}^M \sigma(x_i) \right)^2 \right], \tag{3}$$

where $m$ denotes the number of neurons of the student network and $\theta = (\mathbf{w}_1, \mathbf{w}_2, \ldots, \mathbf{w}_m)^T = (w_{i,j}) \in \mathbb{R}^{m \times d}$. Using this notation, each row of $W$ represents a student neuron.

Assumption 5 allows us to obtain the following analytic characterization of the global minima manifold. This characterization will play a critical role in our theoretical analysis and might be of independent interest to other related problems as well.

**Theorem 6.** *Suppose that $m \geq M$ and Assumption 5 hold. Let $S_0 = \{(0, \ldots, 0) \in \mathbb{R}^d\}$, $S_j = \{\alpha \mathbf{e}_j : \alpha \neq 0\}$ for $j \in [M]$, and $S = \cup_{j=0}^M S_j$. Then the global minima manifold $\mathcal{M}$ is a compact set in $\mathbb{R}^{m \times d}$:*

$$\mathcal{M} = \left\{ \theta = (\mathbf{w}_1, \ldots, \mathbf{w}_m)^T \in \mathbb{R}^{m \times d} : \forall i \in [m], \mathbf{w}_i \in S \text{ and } \forall j \in [M], \sum_{i=1}^m w_{i,j} = 1 \right\} \tag{4}$$

The proof is deferred to Appendix A.1. Note that $S_j \cap S_k = \emptyset$ for any $j \neq k \in \{0, 1, \ldots, M\}$. Hence Theorem 6 implies the following facts about the global minima:

- There are at most $m+1$ types of student neurons, represented by $S_0, S_1, \ldots, S_M$, regardless of how overparameterized the student network is. Moreover, for any $j \in [M]$, there exists at least one student neuron taking the type of $S_j$.

- For each neuron, there exists at most one coordinate to be nonzero and the coordinates from $M+1$ to $d$ must be zero.

The following lemma provides a precise condition of when two global minima are linearly connected, which will be used in our subsequent analysis.

**Lemma 7** (Linear connectivity). *For any two global minima* $\theta^{(1)} = (\mathbf{w}_1^{(1)}, \ldots, \mathbf{w}_m^{(1)})^T$, $\theta^{(2)} = (\mathbf{w}_1^{(2)}, \ldots, \mathbf{w}_m^{(2)})^T \in \mathbb{R}^{m \times d}$, *we have* $W^{(1)} \leftrightarrow W^{(2)}$ *if and only for any* $i \in \{1, \ldots, m\}$, *one of the following happens:*

- $\mathbf{w}_i^{(1)} \in S_0$ *or* $\mathbf{w}_i^{(2)} \in S_0$;

- *there exists* $j \in \{1, \ldots, M\}$ *such that* $\mathbf{w}_i^{(1)} \in S_j$ *and* $\mathbf{w}_i^{(2)} \in S_j$.

The above lemma (proof deferred to Appendix A.1) means that if $\theta_1 \leftrightarrow \theta_2$, then for any $i \in [m]$, the nonzero coordinates of $\mathbf{w}_i^{(1)}$ and $\mathbf{w}_i^{(2)}$ must be the same if they are not zero simultaneously.

### 3.1 The $k$-piece linear connectivity

**Theorem 8.** *Suppose* $m > M$ *and two minima* $\theta^{(1)}, \theta^{(2)}$ *are i.i.d. drawn from* $\mathrm{Unif}(\mathcal{M})$. *Then, w.p. at least* $p_{m,M} = 1 - M(\frac{M^2-1}{M^2})^{m-2M}$, $\theta^{(1)}$ *and* $\theta^{(2)}$ *are 2-PL connected.*

The proof of this theorem can be found in Appendix A.2. Notice that the probability $p_{m,M} \to 1$ as $m \to \infty$, implying that when the student is sufficiently over-parameterized, the 2-PL connectivity holds with probability nearly 1. Quantitatively speaking, for any $\delta \in (0,1)$, $m \geq CM^2 \log(M/\delta)$ is sufficient to guarantee that the probability of 2-PL connectivity is no less than $1 - \delta$.

The following theorem further shows that if allowing the number of pieces to be slightly larger, then the $k$-PL connectivity holds for two global minima.

**Theorem 9.** *Suppose* $m \geq 2M - 1$, *then any two global minima are 4-PL connected.*

The proof is deferred to Appendix A.3.

### 3.2 Star-shaped connectivity

**Theorem 10.** *Suppose* $m > M$ *and let* $\theta_1, \theta_2$ *be two minima i.i.d. drawn from* $\mathrm{Unif}(\mathcal{M})$. *Then, w.p. at least* $1 - M(\frac{M^k-1}{M^k})^{m-kM}$, *there exists a center* $\theta^* \in \mathcal{M}$ *such that* $\theta^i \leftrightarrow \theta^*$ *for all* $i \in [k]$.

A simple calculation implies that to ensure the probability larger than $1 - \delta$, we need $m \geq CM^k \log(M/\delta)$. The following theorem shows by allowing the connectivity between feet and the center to be a two-piece linear path, then the probability becomes exactly 1 as long as $m \geq kM$.

**Theorem 11.** *Given* $k \in \mathbb{N}$, *suppose* $m \geq kM$. *For any* $k$ *global minima* $\theta_1, \ldots, \theta_k$, *we can find a center* $\theta^*$ *such that* $\theta^*$ *and* $\theta_i$ *are 2-PL connected for any* $i = 1, \ldots, k$.

The proofs of Theorem 10 and Theorem 11 can be found in Appendix A.4 and A.5, respectively. In Figure 2, we provide an illustration of the difference in how the feet are connected to the star center between Theorem 10 and Theorem 11.

### 3.3 The geodesic connectivity

The left of Figure 3 shows the normalized geodesic distances (NGDs) between minima found by SGD for the two-layer ReLU networks mentioned above. We can see clearly that the value of NGD decreases monotonically towards 1 as the network width $m$ increases. This implies that the landscape of wide networks should

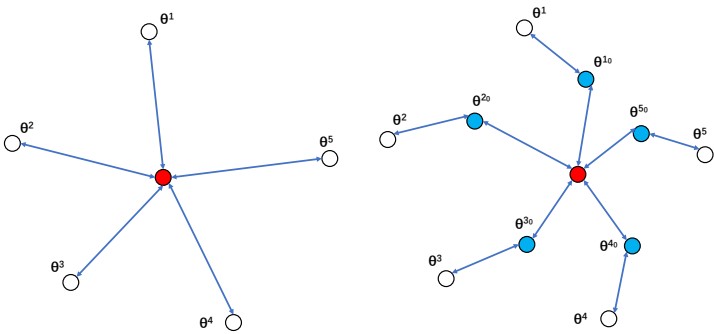

**Figure 2: Left.** The original star-shaped connectivity. The five white circles are the feet and the red circle is the center. The blue line represents the linear connecting path. **Right.** The extended star-shaped connectivity is proved in Theorem 11, where the feet are connected to the center via a two-piece linear path.

be somehow not far from a convex one. This is consistent with our intuition that wider networks should have a simpler landscape than narrow networks. Below, we provide some theoretical evidence to explain this mysterious phenomenon.

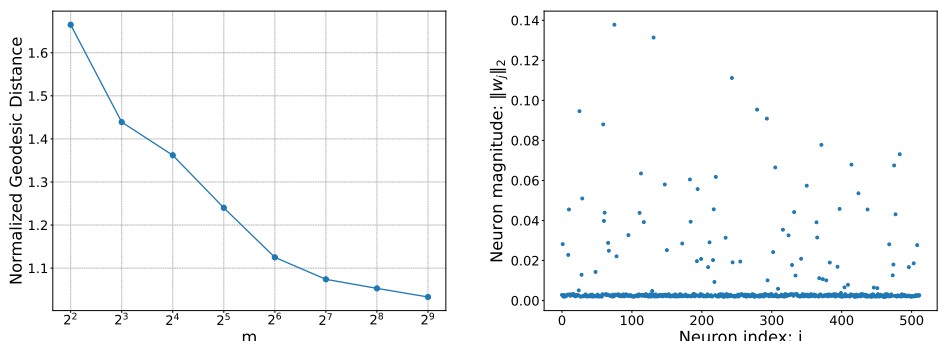

**Figure 3: Left.** How the normalized geodesic distance (NGD) changes with the network width for two-layer ReLU networks. The teacher network has $M = 4$ neurons and we refer to Section 5 for the algorithm of estimating NGD. **Right.** The $L^2$ norm of each neuron for SGD solutions. Here, $m = 512$, $M = 4$, $d = 4$. One can see that SGD tends to find sparse solutions.

**Theorem 12.** *Suppose $m > M$, and let $\theta_1, \theta_2$ be two minima independently drawn from $\mathrm{Unif}(\mathcal{M})$. Then there exists absolute constants $c_1, c_2 > 0$ such that w.p. at least $1 - c_1 e^{-m}$ that $G(\theta_1, \theta_2) \leq c_2\sqrt{M}$. Moreover, the upper bound can be achieved by a two-piece linear path.*

The proof is deferred to Appendix A.6. This theorem shows that the NGD between uniformly sampled minima is independent of the level of over-parameterization. However, Figure 3 shows that NGD shrinks to 1 when increasing the network width for minima found by SGD. We hypothesize that SGD induces a biased distribution over the minima manifold. In the right of Figure 3, we visualize the magnitude of different neurons for an SGD solution. We observe that SGD tends to find solutions with sparse structures, i.e., only a few dominant neurons contribute.

To study the influence of neuron sparsity on the geodesic connectivity, we define the following distribution to formulate the sparsity bias.

**Definition 13** (Neuron-sparse distribution)**.** For any absolute constant $0 < r < 1$, we define $\mathrm{SP}(\mathcal{M}, r)$, the neuron-sparse distribution with a sparsity $r$ over $\mathcal{M}$, as following: for $\theta = (\mathbf{w}_1, \ldots, \mathbf{w}_m)^T$, for any $i \in \{1, \ldots, m\}$, $P(\mathbf{w}_i \in S_0) = r$ and $P(\mathbf{w}_i \in S_j) = \frac{1-r}{M}$ for $j \in \{1, \ldots, M\}$.

**Theorem 14.** *Suppose $m > M$ and let $\theta_1, \theta_2$ be two minima independently drawn from $\mathrm{SP}(\mathcal{M}, r)$. Then there exists two absolute constants $c_1, c_2 > 0$ such that w.p.at least $1 - c_1 M e^{-mr^2}$ that $G(\theta_1, \theta_2) \leq 1 + \frac{c_2}{r\sqrt{m}}$. Moreover, the upper bound can be achieved by a two-piece linear path.*

The proof is deferred to Appendix A.7. This theorem demonstrates that the bias towards sparse solutions can explain the shrinkage of NGD to 1. In particular, when $m \to \infty$, NGD approaches to 1.

# 4 Linear networks

A linear network $f(\cdot; \theta) : \mathbb{R}^d \mapsto \mathbb{R}$ is parameterized by $f(\mathbf{x}; \theta) = A_L A_{L-1} \ldots A_1 \mathbf{x}$, where $A_l \in \mathbb{R}^{m_l \times m_{l-1}}$ for $l = 1, 2, \ldots, L$. Here $L$ denotes the network depth and $\{m_l\}_{l=0}^L$ denotes the widths. Note that $m_0 = d$ and $m_L = 1$ and we assume $m_2 = \cdots = m_{L-1} = m$ for simplicity. We make the following assumption on the data distribution.

**Assumption 15.** Suppose that $y = Q\mathbf{x}$ for some $Q \in \mathbb{R}^{1 \times d}$, $\mathbb{E}[\mathbf{x}] = 0$, and $\lambda_{\min}(\mathbb{E}[\mathbf{x}\mathbf{x}^T]) > 0$.

The above assumption is quite mild but ensures that the global minima manifold has the following analytic characterization:

$$\mathcal{M} = \{(A_L, A_{L-1}, \ldots, A_1) : A_L A_{L-1} \cdots A_1 = Q\} \tag{5}$$

The following theorem provides the characterization of $k$-PL connectivity of the loss landscape of linear networks. The proof can be found in Appendix B.

**Theorem 16.** *Let $f(\cdot; \theta)$ be the linear network described in Section 4. Let $\theta_1, \theta_2 \in \mathcal{M}$ be two global minima. If $m > 2L - 1$, then we have:*

- *Two global minima are almost surely 2-PL connected;*

- *Any two global minima are 3-PL connected,*

We remark that the "almost surely" condition in characterizing the 2-PL connectivity cannot be removed. The following lemma provides a counterexample, showing that there indeed exist pathological minima that are not 2-PL connected.

**Lemma 17.** *Consider the case of $m = 4, d = 1, L = 2$ and the target $y = x$. Then, we have $\mathcal{M} = \{(a, b) \in \mathbb{R}^m \otimes \mathbb{R}^m : a^T b = 1\}$. Consider two global minima $\theta_1 = (A_1^{(1)}, A_1^{(2)})$, $\theta_2 = (A_2^{(1)}, A_2^{(2)})$ with*

$$A_1^{(1)} = (1, 0, 0, 0), A_2^{(1)} = (1, 0, 0, 0)^T, \quad A_1^{(2)} = (-1, 0, 0, 0), A_2^{(2)} = (-1, 0, 0, 0)^T.$$

*Then, $\theta_1$ and $\theta_2$ are not 2-PL connected. Quantitatively,*

$$\inf_{\theta \in \mathcal{M}} \left( \int_0^1 \mathcal{R}(t\theta_1 + (1-t)\theta) \, \mathrm{d}t + \int_0^1 \mathcal{R}(t\theta_2 + (1-t)\theta) \right) \mathrm{d}t \geq \frac{4}{15} \mathbb{E}x^2.$$

**Theorem 18** (star-shaped linear connectivity)**.** *Consider linear networks of depth $L$ and width $m$. Let $\{\theta_i\}_{i=1}^r$ be $r$ global minima. If $m > 1 + r(L-1)$, then we almost surely (with respect to the Lebesgue measure over $\mathcal{M}^{\otimes r}$) have that there exists a $\theta^* \in \mathcal{M}$ such that $\theta^* \leftrightarrow \theta_i$ for all $i = 1, \ldots, r$.*

Here $\mathcal{M}^{\otimes r} = \{(\theta_1, \ldots, \theta_r) \in \mathbb{R}^r : \theta_i \in \mathcal{M} \text{ for } i \in [r]\}$. This theorem establishes that when linear networks are sufficiently wide, the star-shaped connectivity holds almost surely. The proof can be found in Appendix B.1.

**The geodesic connectivity.** In Figure 4, we empirically demonstrate that the Normalized Geodesic Distances (NGDs) for linear networks are also close to 1. Additionally, we observe that the NGD monotonically decreases with increasing network width, although this phenomenon has not yet been theoretically proven.

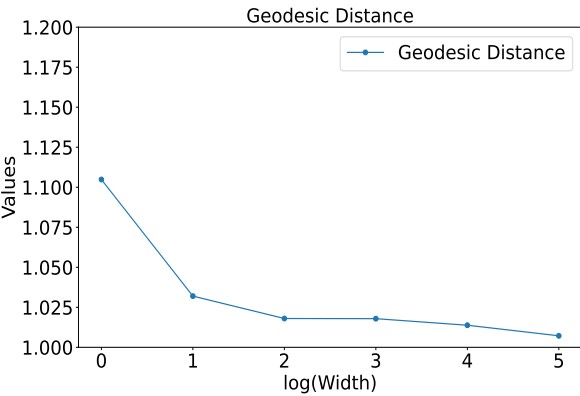

**Figure 4:** Normalized geodesic distance vs. network width for linear networks. Following the setting as described earlier in this section, we consider a fully connected linear network with $L = 2$. We set $d = m$, and vary $m$ to consider the normalized geodesic distance of a center with 2-PL-connectivity. Algorithm 5 is applied here to train a center and the result is an average of 5 separate experiments. It is shown that as the width increases, we can obtain a center that satisfies 2-PL-connectivity with a shorter geodesic distance.

## 5  Experiments

In this section, we provide experiments to validate the star-shaped and geodesic connectivity across a range of architectures and datasets.

**The center-finding algorithm.**  Given a set of minima $S = \{\theta_i^*\}_{i=1}^r$, to find a center $\theta$ that connects to all of them via linear paths, we propose to minimize the following objective

$$\mathcal{J}_S(\theta) := \frac{1}{r} \sum_{i=1}^r \left( \mathbb{E}_{t \sim U[0,1]}[\mathcal{R}(t\theta + (1-t)\theta_i^*)] + \lambda p(\theta, \theta_i^*) \right), \tag{6}$$

where $p(\cdot, \cdot)$ is a penalization function to be determined later. To efficiently solve this optimization problem, we use the Adam Kingma & Ba (2014) optimizer with the minibatch gradient:

$$\nabla \widehat{\mathcal{J}}_S(\theta) = \nabla \left( \frac{1}{B_r B_t} \sum_{k=1}^{B_r} \sum_{j=1}^{B_t} \mathcal{R}(t_j \theta_t + (1 - t_j)\theta_{i_k}^*) + \frac{\lambda}{B_r} \sum_{k=1}^{B_r} p(\theta_t, \theta_{i_k}^*) \right), \tag{7}$$

where $i_k \overset{\text{i.i.d.}}{\sim} \text{Unif}([r])$ and $t_j \overset{\text{i.i.d.}}{\sim} \text{Unif}([0,1])$ for $k \in [B_r]$ and $j \in [B_t]$. Here, $B_r, B_t \in \mathbb{N}$ denote the batch sizes. Across all our experiments, we always set $B_r = 1, B_t = 3$.

**Estimating the normalized geodesic distance.**  Given two minima $\theta_1^*$ and $\theta_2^*$, we first find a center $\tilde{\theta} \in \mathcal{M}$ by minimizing the objective equation 6 for $S = \{\theta_1^*, \theta_2^*\}$ and $p(\theta, \theta') = \|\theta - \theta'\|_2^2$. This allows us to find a minimum on the minima manifold such that it connects to both $\theta_1^*$ and $\theta_2^*$ via the shortest linear paths. Moreover, by Definition 4, it holds for any $\theta \in \mathcal{M}$, satisfying $\theta \leftrightarrow \theta_i^*$ for $i = 1, 2$, that

$$G(\theta_1^*, \theta_2^*) \leq \frac{\|\theta_1^* - \theta\|_2 + \|\theta - \theta_2^*\|_2}{\|\theta_1^* - \theta_2^*\|_2}. \tag{8}$$

Then, plugging $\tilde{\theta}$ into the right hand side of equation 8 gives an upper bound of $G(\theta_1^*, \theta_2^*)$.

**The experiment setup.**  To validate our theoretical findings for practical models, we train fully-connected neural networks (FNNs) and VGG16 Simonyan & Zisserman (2014) for classifying MNIST LeCun et al. (1998) dataset, and VGG16 Simonyan & Zisserman (2014) and ResNet34 He et al. (2016) for classifying CIFAR-10 Krizhevsky et al. (2009) dataset, respectively:

- The FNN is three-layer, whose architecture is $728 \to 500 \to 300 \to 10$. We trained FNNs under the hyperparameters: lr = 1e-3 and batchsize = 200,

- The architecture of VGG16 and ResNet34 can be found in Simonyan & Zisserman (2014) and He et al. (2016), respectively. We trained VGG16 and ResNet34 under the hyperparameters: lr = 5e-3 and batchsize = 200.

As for the center-finding algorithm, we set $B_r = 1$, $B_t = 3$ as mentioned above, and learning rate $\eta = 0.01$. For all cases, the Adam optimizer Kingma & Ba (2014) is adopted.

## 5.1 Star-shaped connectivity

In Figure 5, we visualize the star-shaped connectivity for VGG-16 in classifying the CIFAR10 dataset. We independently train the model to find 3 minima $\{\theta_i^*\}_{i=1}^3$ and then run the center-finding algorithm to locate a center $\theta_{c^*}$ on the minima manifold. We can see that the linear interpolation between any pair minima among the three ones indeed encounters a very high barrier. However, through the center $\theta_c$, the three minima form a star-shaped connectivity.

In Table 1, we provide more experiments for a variety of model architectures and training datasets. We can see that the star-shaped connectivity holds for all scenarios examined.

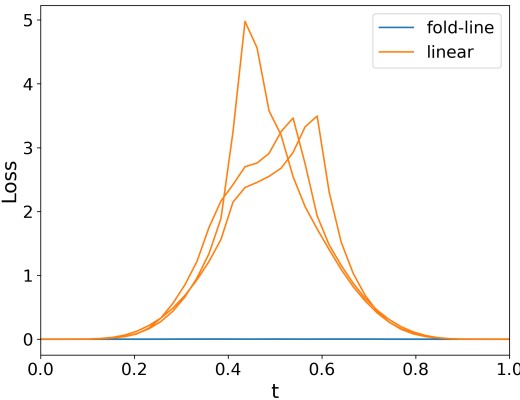 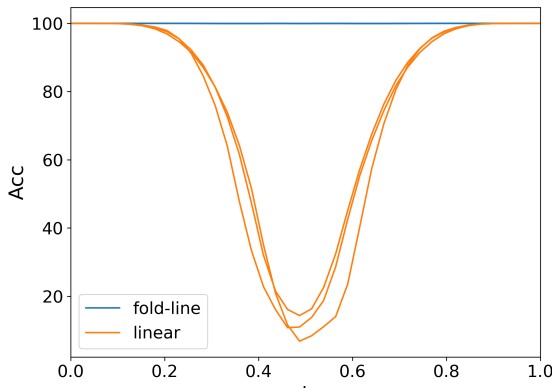

**Figure 5:** An validation of star-shaped connectivity. The model is VGG16 and the dataset is CIFAR-10. We examine 3 minima obtained by running Adam independently. Then we applied the center-finding algorithm to obtain the corresponding center. For all the 3 linear interpolations between minima, and all the 3 "minimum-center-minimum" fold-lines constructed by two linear interpolations, we plot the training loss (left) and accuracy (right) along these paths. Specifically, the $x$-axis $t$ here denotes the point $t\theta^* + (1-t)\theta_i$ in the linear interpolation (the orange line). On the other hand, for a pair $(\theta_i, \theta_j)$ (blue line), $t < 0.5$ corresponds to the point $2t\theta^* + (1-2t)\theta_i$, while $t \geq 0.5$ corresponds to $(2t-1)\theta_j + (2-2t)\theta^*$. It is shown in the experiment that our algorithm successfully found a center that is linearly connected to all three minima simultaneously, i.e., forms a star-shaped connectivity.

## 5.2 The geodesic connectivity

In Table 2, we report the NGDs estimated by the aforementioned algorithm. It is demonstrated that minima (found by the commonly-used Adam optimizers) can be connected via paths whose NGDs are nearly 1. This is consistent with our theoretical findings in toy models.

|  | MNIST | | CIFAR10 | |
|---|---|---|---|---|
|  | VGG16 | FNN | VGG16 | ResNet34 |
| Loss Barrier (linear) | 16.91 | 1.25 | 6.21 | 3.28 |
| Loss Barrier (fold-line) | 3.1e-05 | 1.1e-03 | 5.0e-03 | 1.0e-02 |

|  | MNIST | | CIFAR10 | |
|---|---|---|---|---|
|  | VGG16 | FNN | VGG16 | ResNet34 |
| Accuracy Barrier (linear) | 28.55% | 68.09% | 10.79% | 41.21% |
| Accuracy Barrier (fold-line) | 100.00% | 99.96% | 99.99% | 99.65% |

**Table 1:** For different models and datasets, we independently trained 5 minima using Adam optimizer. Then, we run the path-finding algorithm with $B_r = 1, B_t = 3$ for 200 epochs. For all the 10 linear interpolations between minima, and all the 10 "minimum-center-minimum" fold-lines constructed by two linear interpolations, we computed the maximum (for loss) or minimum (for accuracy) and averaged them, which is denoted as "barrier". It is shown that there is nearly no barrier on the fold-lines we constructed, which validates that our observation of star-shaped connectivity holds for a wide range of settings.

|  | FNN+MNIST | VGG16+MNIST | VGG16+CIFAR10 | ResNet18+CIFAR10 |
|---|---|---|---|---|
| NGD | 1.003 | 1.001 | 1.051 | 1.003 |
| Barrier | 99.89% | 99.25% | 99.91% | 99.63% |

**Table 2:** The upper bound of NGD estimated via Eq. 8 for different networks and datasets. We independently trained 2 minima in each setting, then trained the center via the center-finding algorithm with a penalized term. It turns out that we can obtain great linear mode connectivity via a fold-line, as well as keeping the relative distance proportion controlled.

## 6 Conclusion

In this paper, we systematically investigate the star-shaped and geodesic connectivity phenomenon for the landscape of neural networks. We provide theoretical analysis on toy models such as two-layer ReLU networks and linear networks, as well as experimental validations on popular networks trained on MNIST and CIFAR-10 datasets. Our findings reveal that the neural network landscape has many simple structures. Specifically, the star-shaped phenomenon suggests a connectivity property stronger than mode connectivity. The geodesic connectivity indicates that the loss landscape might be not far from being convex in a certain sense. For future work, it would be interesting to explore the potential relationships between our findings and optimization and generalization in neural networks.

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

# A  Proofs in Section 3

## A.1  Proof of Theorem 6.

Let

$$\widetilde{\mathcal{M}} = \left\{ W \in \mathbb{R}^{m \times d} : \mathbf{w}_i \in S \text{ for } i = 1, \ldots, m \text{ and } \sum_{i=1}^{m} w_{i,j} = 1 \text{ for } j = 1, \ldots, M \right\}.$$

Our task is to prove that $\mathcal{M} = \widetilde{\mathcal{M}}$. For any $W \in \mathcal{M}$, we have $L(W) = \mathbb{E}_{\mathbf{x} \sim \tau_{d-1}}[(\sum_{i=1}^{m} \sigma(\mathbf{w}_i^T \mathbf{x}) - \sum_{j=1}^{M} \sigma(x_j))^2] = 0$. By the non-degeneracy of $\tau_{d-1}$, this is equivalent to

$$\sum_{i=1}^{m} \sigma(\mathbf{w}_i^T \mathbf{x}) = \sum_{j=1}^{M} \sigma(x_j) \quad \forall \mathbf{x} \in \mathbb{S}^{d-1}. \tag{9}$$

- We first consider the first $M$ columns. Taking $\mathbf{x} = \mathbf{e}_j$ in equation 9 gives for any $j \in [M]$ that

$$\sum_{i=1}^{m} \sigma(w_{i,j}) = \sum_{i=1}^{m} \sigma(\mathbf{w}_i^T \mathbf{e}_j) = 1. \tag{10}$$

  Taking $\mathbf{x} = -\mathbf{e}_j$ in equation 9 gives for any $j \in [M]$ that

$$\sum_{i=1}^{m} \sigma(-w_{ij}) = \sum_{i=1}^{m} \sigma(-\mathbf{w}_i^T \mathbf{e}_j) = 0. \tag{11}$$

  Combining equation 10 and equation 11 leads to

$$\begin{cases} w_{ij} \geq 0 & \forall i \in [m], j \in [M] \\ \sum_{i=1}^{m} w_{ij} = 1 & \forall j \in [M] \end{cases}. \tag{12}$$

- Next we turn to consider the columns from $M+1$ to $d$. Analogously, for any $j \in M+1, \ldots, d$, we take $\mathbf{x} = \mathbf{e}_j$ and $\mathbf{x} = -\mathbf{e}_j$ in equation 9, yielding

$$\begin{cases} \sum_{i=1}^{m} \sigma(w_{ij}) = 0 \\ \sum_{i=1}^{m} \sigma(-w_{ij}) = 0 \end{cases}. \tag{13}$$

  This implies

$$w_{ij} = 0, \quad \forall i \in \{1, \ldots, m\}, j \in \{M+1, \ldots, d\}. \tag{14}$$

- Now we prove by contradiction that for each $j \in [m]$, there exists at most one coordinate to be nonzero. Suppose that there exists $i \in [m]$ such that $\|w_i\|_0 \geq 2$. W.L.O.G, let $i = 1$ and $w_{11} > 0, w_{12} > 0$. Then, there must exist $\epsilon > 0$ such that $\sqrt{1 - \epsilon^2} w_{11} - \epsilon w_{12} > 0$.

  Let $\mathbf{x} = \sqrt{1 - \epsilon^2} \mathbf{e}_1 - \epsilon \mathbf{e}_2$ in equation 9. First, we have $\sum_{i=1}^{M} \sigma(x_i) = \sqrt{1 - \epsilon^2}$. Second,

$$\sum_{i=1}^{m} \sigma(\mathbf{w}_i \cdot \mathbf{x}) = \sum_{i=1}^{m} \sigma(\sqrt{1 - \epsilon^2} w_{i1} - \epsilon w_{i2}) = \sigma(\sqrt{1 - \epsilon^2} w_{11} - \epsilon w_{12}) + \sum_{i=2}^{m} \sigma(\sqrt{1 - \epsilon^2} w_{i1} - \epsilon w_{i2})$$

$$\leq \sigma(\sqrt{1 - \epsilon^2} w_{11} - \epsilon w_{12}) + \sum_{i=2}^{m} \sigma(\sqrt{1 - \epsilon^2} w_{i1}) = \sqrt{1 - \epsilon^2} w_{11} - \epsilon w_{12} + \sum_{i=2}^{m} \sqrt{1 - \epsilon^2} w_{i1}$$

$$= \sqrt{1 - \epsilon^2} \sum_{i=1}^{m} w_{i1} - \epsilon w_{i2} < \sqrt{1 - \epsilon^2} \sum_{i=1}^{m} w_{i1} = \sqrt{1 - \epsilon^2}.$$

  Thus, $\sum_{i=1}^{m} \sigma(\mathbf{w}_i \cdot \mathbf{x}) < \sum_{i=1}^{M} \sigma(x_i)$, which is contradictory with equation 9.

Combining the three conclusions above, we proved $\mathcal{M} \subset \widetilde{\mathcal{M}}$. What remains is to prove $\widetilde{\mathcal{M}} \subset \mathcal{M}$. It is obvious that for any $W \in \widetilde{\mathcal{M}}$, we have $\sum_{i=1}^{m} \sigma(\mathbf{w}_i \cdot \mathbf{x}) = \sum_{i=1}^{m} \sigma\left(\sum_{j=1}^{d} w_{ij} x_j\right) = \sum_{i=1}^{m} \sum_{j=1}^{d} w_{ij} \sigma(x_j) = \sum_{i=1}^{M} \sigma(x_i)$. Thus $W$ is a global minimum, implying $\tilde{\mathcal{M}} \subset \mathcal{M}$. $\qquad \square$

**Proof of Lemma 7.** Recall that Theorem 6 shows

$$\mathcal{M} = \left\{ W \in \mathbb{R}^{m \times d} : \mathbf{w}_i \in S \text{ for } i = 1, \ldots, m \text{ and } \sum_{i=1}^{m} w_{i,j} = 1 \text{ for } j = 1, \ldots, M \right\}, \tag{15}$$

where $S = \cup_{j=0}^{M} S_j$ with $S_0 = \{(0, \ldots, 0) \in \mathbb{R}^d\}$ and $S_j = \{\alpha \mathbf{e}_j \in \mathbb{R}^d : \alpha \neq 0\}$ for $j = 1, \ldots, M$.

Given any $\theta_1, \theta_2 \in \mathcal{M}$, our task is to prove that $\theta_1 \leftrightarrow \theta_2$ is equivalent to that one of the following two conditions is satisfied:

a) $\mathbf{w}_i^{(1)} \in S_0$ or $\mathbf{w}_i^{(2)} \in S_0$;

b) there exists $j \in \{1, \ldots, M\}$ such that $\mathbf{w}_i^{(1)} \in S_j$ and $\mathbf{w}_i^{(2)} \in S_j$.

We first prove that $\theta_1 \leftrightarrow \theta_2$ can lead to the condition a) or b). Note that $\theta_1 \leftrightarrow \theta_2$ means

$$\gamma(t) = ((1-t)\mathbf{w}_1^{(1)} + t\mathbf{w}_1^{(2)}, \cdots, (1-t)\mathbf{w}_m^{(1)} + t\mathbf{w}_m^{(2)})^T \in \mathcal{M}.$$

By equation 15, $(1-t)\mathbf{w}_i^{(1)} + t\mathbf{w}_i^{(2)} \in S$ holds for any $i \in [m]$ and $t \in [0,1]$. Since any element in $S$ has at most one nonzero coordinate, $\mathbf{w}_i^{(1)}$ and $\mathbf{w}_i^{(2)}$ have at most one nonzero coordinate and their nonzero coordinates must be the same. Otherwise, the number of nonzero coordinates of $(1-t)\mathbf{w}_i^{(1)} + t\mathbf{w}_i^{(2)}$ will be no less than 2 for any $t \in (0,1)$. This implies that either condition a) or condition b) is satisfied.

Second, if condition a) or condition b) is satisfied, then for any $t \in [0,1]$ and any $i \in [m]$, $(1-t)\mathbf{w}_i^{(1)} + t\mathbf{w}_i^{(2)} \in S$. Moreover, for any $j \in [M]$,

$$\sum_{i=1}^{m} \left((1-t)w_{i,j}^{(1)} + tw_{i,j}^{(2)}\right) = (1-t)\sum_{i=1}^{m} w_{i,j}^{(1)} + t\sum_{i=1}^{m} w_{i,j}^{(2)} = (1-t) + t = 1.$$

Hence, by equation 15, $(1-t)\theta_1 + t\theta_2 \in \mathcal{M}$ for any $t \in [0,1]$. $\qquad\square$

## A.2 Proof of Theorem 8.

Let $\theta_i \overset{iid}{\sim} \text{Unif}(\mathcal{M})$ for $i = 1, 2$. We aim to give a lower bound for the probability that there exists a global minimum $\theta^*$ such that $\theta^1 \leftrightarrow \theta^*$ and $\theta^* \leftrightarrow \theta^2$. From Theorem 6, given a $\theta = (\mathbf{w}_1, \ldots, \mathbf{w}_m)^T \in \mathbb{R}^{m \times d}$, the sufficient and necessary condition for $\theta$ to be a global minimum is:

(1) $\mathbf{w}_i \in S$, for any $i \in [m]$.

(2) $\sum_{j:\mathbf{w}_j \in S_i} \mathbf{w}_j = e_i$, for $i \in \{1, \ldots, M\}$.

We notice that $\theta^1(\theta^2)$ has the requirement that for every $i \in \{1, \ldots, M\}$, there exists $j \in \{1, \ldots, m\}$ such that $\theta_j^1(\theta_j^2) \in S_i$, after we already have $M$ different elements of $\theta^1(\theta^2)$, the rest elements have no restriction and can be arbitrarily chosen in at least $m - 2M$ overlapped positions.

Hence, we suppose $\mathbf{w}_j^1$ and $\mathbf{w}_j^2$ have uniform distribution over $S$ for any $j \in T$, where $T$ is a subset of $\{1, \ldots, m\}$ containing $m - 2M$ elements. i.e. $\mathbf{w}_j^1(\mathbf{w}_j^2)$ $(j \in T)$ chooses randomly a set from $\{S_0, \ldots, S_M\}$ to belong to.

For $j \in T$, we consider the pair $(\mathbf{w}_j^1, \mathbf{w}_j^2)$. We denote $[\mathbf{w}_j^1, \mathbf{w}_j^2] \overset{\Delta}{=} [p, q]$ if $\mathbf{w}_j^1 \in S_p$ and $\mathbf{w}_j^2 \in S_q$.

Then we define the incident $A_i$: for any $j \in T$, $[\mathbf{w}_j^1, \mathbf{w}_j^2] \neq [i, i]$. Since $\theta_i \overset{iid}{\sim} \text{Unif}(\mathcal{M})$ for $i = 1, 2$, we have $P(A_i) = (\frac{M^2 - 1}{M^2})^{m - 2M}$.

Thus, from the inclusion-exclusion principle, we have:

$$P(\theta_1 \text{ and } \theta_2 \text{ are 2-PL connected})$$

$$\geq 1 - \sum_{i=1}^{M} P(A_i)$$

$$= 1 - MP(A_1)$$

$$= 1 - M\left(\frac{M^2 - 1}{M^2}\right)^{m - 2M}.$$

□

## A.3 Proof of Theorem 9.

First, we consider the choice of $\theta^3$ ($\theta^5$). Since $\theta^1$ ($\theta^2$) has property that for every $i \in \{1, \ldots, M\}$, there exists $j_i \in \{1, \ldots, m\}$ such that $\mathbf{w}_{j_i}^1 \in S_i$. Then, we let $\mathbf{w}_{j_i}^3 = \mathbf{e}_i$ for $i \in \{1, \ldots, M\}$, and set the other line vector of $\theta^3$ as zero. From Theorem 6, $\theta^3$ is a global minimum of Equation (3). Further, from Lemma 7, we have $\theta^1 \leftrightarrow \theta^3$. We call this method generating $\theta^3$ from $\theta^1$ "merging", since it straightforwardly merges some line vectors of $\theta^1$ belonging to the same set in $S$ to a single non-zero vector. Similarly, we can merge $\theta^2$ to $\theta^5$.

$\theta^3$ and $\theta^5$ share the common characteristic that they contain exactly $M$ different non-zero line vectors $\{\mathbf{e}_1, \ldots, \mathbf{e}_M\}$, and $m - M$ zero line vectors. Since $m \geq 2M - 1$, we have $m - M \geq M - 1$, thus $\theta^3$ has at least $M - 1$ zero line vectors.

**Case 1.**

If there are at least $M$ zero line vectors, suppose the set of the zero line vectors is $\{\mathbf{w}_{a_1}^3, \ldots, \mathbf{w}_{a_M}^3\}$. Then, since $\mathbf{w}_{a_1}^5, \ldots, \mathbf{w}_{a_M}^5$ belong to different subsets of $S$ or belong to $S_0$, we can find a feasible set of $\{\mathbf{w}_{a_1}^4, \ldots, \mathbf{w}_{a_M}^4\} = \{\mathbf{e}_1, \ldots, \mathbf{e}_M\}$. We set other line vectors of $\theta^4$ as zero and then we are done since we have a feasible global minimum $\theta^4$.

**Case 2.**

If there are only $M - 1$ zero line vectors for $\theta^3$. Now, we fix a feasible $\theta^3$ merged from $\theta^1$, suppose the set of its zero line vectors is $\{\mathbf{w}_{a_1}^3, \ldots, \mathbf{w}_{a_{M-1}}^3\}$. First, we look at the corresponding line vectors of these $\theta^3$'s zero vectors in $\theta^5$, i.e. $\{\mathbf{w}_{a_1}^5, \ldots, \mathbf{w}_{a_{M-1}}^5\}$. We notice that $\theta^5$ is not fixed when generated from $\theta^2$, and trivially we have at least $M$ different positions to choose from for the $M - 1$ zero line vectors. Hence, there exists a choice of $\theta^5$ where at least one element of $\{\mathbf{w}_{a_1}^5, \ldots, \mathbf{w}_{a_{M-1}}^5\}$ is non-zero. Thus, there is at least one zero vector of $\theta^5$ with a non-zero corresponding line vector in $\theta^3$, the set of which we denote as $\{\mathbf{w}_{b_1}^3, \ldots, \mathbf{w}_{b_f}^3\}$.

**Case 2-1.**

If there exists $\mathbf{w}_{b_i}^3$ belongs to different subset of $S$ from any element of $\{\mathbf{w}_{a_1}^5, \ldots, \mathbf{w}_{a_{M-1}}^5\}$, then we can find a feasible set of $\{\mathbf{w}_{b_i}^4, \mathbf{w}_{a_1}^4, \ldots, \mathbf{w}_{a_{M-1}}^4\} = \{\mathbf{e}_1, \ldots, \mathbf{e}_M\}$. We set other line vectors of $\theta^4$ as zero and then we are done since we have a feasible global minimum $\theta^4$.

**Case 2-2.**

If every element of $\mathbf{w}_{b_1}^3, \ldots, \mathbf{w}_{b_f}^3$ belongs to the same subset of $S$ as an element of $\{\mathbf{w}_{a_1}^5, \ldots, \mathbf{w}_{a_{M-1}}^5\}$, we arbitrarily choose a line vector from $\mathbf{w}_{b_1}^3, \ldots, \mathbf{w}_{b_f}^3$. Suppose $\mathbf{w}_{b_j}^3 \in S_t$ and $\mathbf{w}_{a_q}^5 \in S_t$. Then we suppose $\mathbf{w}_{b_j}^2 \in S_p$, and $\mathbf{w}_{b_g}^5 \in S_p$.

**Case 2-2-1.**

If $p = 0$, then we let $\mathbf{w}_{b_j}^5 = e_t$, and let $\mathbf{w}_{a_q}^5 = 0$. Then we can find a feasible set of $\{\mathbf{w}_{b_j}^4, \mathbf{w}_{a_1}^4, \ldots, \mathbf{w}_{a_{M-1}}^4\} = \{\mathbf{e}_1, \ldots, \mathbf{e}_M\}$. We set other line vectors of $\theta^4$ as zero and then we are done since we have a feasible global minimum $\theta^4$.

**Case 2-2-2.**

If $p \neq 0$, then we can switch $\mathbf{w}_{b_j}^5$ and $\mathbf{w}_{b_g}^5$ without damaging the connectivity of the global minima. Now, the number of the non-zero corresponding line vectors in $\theta^3$ reduces to $f - 1$. After at most $f - 1$ same operations, if we still didn't find the feasible global minimum $W_4$, then we get exactly one non-zero corresponding line vector in $\theta^3$ and exactly one non-zero element in $\{\mathbf{w}_{a_1}^5, \ldots, \mathbf{w}_{a_{M-1}}^5\}$, and further, these two vectors belong to the same subset of $S$. Suppose $\mathbf{w}_{b_{j'}}^3 \in S_{t'}$ and $\mathbf{w}_{a_{q'}}^5 \in S_{t'}$ ($t' \neq 0$). Then we suppose $\mathbf{w}_{b_{j'}}^2 \in S_{p'}$, and $\mathbf{w}_{b_{g'}}^5 \in S_{p'}$.

**Case 2-2-2-1.**

If $p' = 0$, then we let $\mathbf{w}_{b_{j'}}^5 = e_{t'}$, and let $\mathbf{w}_{a_{q'}}^5 = 0$. Then we can find a feasible set of $\{\mathbf{w}_{b_{j'}}^4, \mathbf{w}_{a_1}^4, \ldots, \mathbf{w}_{a_{M-1}}^4\} = \{\mathbf{e}_1, \ldots, \mathbf{e}_M\}$. We set other line vectors of $\theta^4$ as zero and then we are done since we have a feasible global minimum $\theta^4$.

**Case 2-2-2-2.**

If $p' \neq 0$, then we can switch $\mathbf{w}_{b_{j'}}^5$ and $\mathbf{w}_{b_{g'}}^5$ without damaging the connectivity of the global minima. Since $p' \neq t'$, we can find a feasible set of $\{\mathbf{w}_{b_{g'}}^4, \mathbf{w}_{a_1}^4, \ldots, \mathbf{w}_{a_{M-1}}^4\} = \{\boldsymbol{e}_1, \ldots, \boldsymbol{e}_M\}$. We set other line vectors of $\theta^4$ as zero and then we are done since we have a feasible global minimum $\theta^4$.

Concluding above, we completed the proof. $\qquad\square$

## A.4 Proof of Theorem 10

We follow the proof of Theorem 8 and apply some modifications.

Here, for $j \in T$, we consider the combination $(\mathbf{w}_j^1, \ldots, \mathbf{w}_j^k)$. We denote $[\mathbf{w}_j^1, \ldots \mathbf{w}_j^k] \overset{\Delta}{=} [p_1, \ldots, p_k]$ if $\mathbf{w}_j^i \in S_{p_i}$ for $i \in \{1, \ldots, k\}$.

Similarly, we define the incident $A_i$: for any $j \in T$, $[\mathbf{w}_j^1, \mathbf{w}_j^2, \ldots, \mathbf{w}_j^k] \neq [i, i, \ldots, i]$.

Thus. following the proof of Theorem 8, based on our assumption for uniform distribution, we can calculate that $P(A_i) = (\frac{M^k - 1}{M^k})^{m - kM}$.

Hence, $1 - \sum_{i=1}^{M} P(A_i) = 1 - M(\frac{M^k - 1}{M^k})^{m - kM}$, which is the lower bound in the theorem. $\qquad\square$

## A.5 Proof of Theorem 11.

We follow the proof of Theorem 9. Firstly, we merge $\theta^i$ to be $\theta^{i_0}$ just as what we did in the proof of Theorem 9. Considering $\theta^{i_0}$ for any $i \in \{1, \ldots, k\}$, since $m \geq kM$, it has at least $(k-1)M$ zero line vectors.

Hence, it trivially holds that $\theta^{1_0}$ and $\theta^{2_0}$ share at least $(k-2)M$ zero line vectors of relatively same position, $\theta^{1_0}$, $\theta^{2_0}$ and $\theta^{3_0}$ share at least $(k-3)M$ zero line vectors of relatively same position... We can easily use mathematical induction to deduce that $\theta^{1_0}, \ldots, \theta^{k-1_0}$ share at least $M$ zero line vectors of relatively same position. Suppose the positions of the $M$ common line vectors are $\{a_1, \ldots, a_M\}$. Since $\mathbf{w}_{a_1}^k, \ldots, \mathbf{w}_{a_M}^k$ belong to different subsets of $S$ or belong to $S_0$, we can find a feasible set of $\{\mathbf{w}_{a_1}^{k+1}, \ldots, \mathbf{w}_{a_M}^{k+1}\} = \{\boldsymbol{e}_1, \ldots, \boldsymbol{e}_M\}$. We set other line vectors of $\theta^{k+1}$ as zero and then we are done since we have a feasible global minimum $\theta^{k+1}$.

$\qquad\square$

## A.6 Proof of Theorem 12

**Lemma 19.** *For any two global minima $\theta_1 = (\boldsymbol{u}_1^1, \ldots, \boldsymbol{u}_d^1), \theta_2 = (\boldsymbol{u}_1^2, \ldots, \boldsymbol{u}_d^2)$, for $i \in \{1, \ldots, M\}$, suppose $\theta_1$ and $\theta_2$ respectively have $m_1$ and $m_2$ neurons that belong to $S_i$, with $m_t$ neurons sharing common coordinates and $m_t \leq \min\{m_1, m_2\}$. Then, for any global minimum $\theta = (\boldsymbol{v}_1, \ldots, \boldsymbol{v}_d)$, we have*

$$\min_{\theta} \frac{\|\boldsymbol{v}_i - \boldsymbol{u}_i^1\|_2^2 + \|\boldsymbol{v}_i - \boldsymbol{u}_i^2\|_2^2}{\|\boldsymbol{u}_i^1 - \boldsymbol{u}_i^2\|_2^2} \leq \frac{1}{1 - \sqrt{\frac{(m_1 - m_t)(m_2 - m_t)}{m_1 m_2}}}.$$

**Proof of Lemma 19.** Suppose $\theta_1 = (\mathbf{w}_1^1, \ldots, \mathbf{w}_m^1)^T = (\boldsymbol{u}_1^1, \ldots, \boldsymbol{u}_d^1), \theta_2 = (\mathbf{w}_1^2, \ldots, \mathbf{w}_m^2)^T = (\boldsymbol{u}_1^2, \ldots, \boldsymbol{u}_d^2) \in \mathbb{R}^{m \times d}$. Suppose $\theta = (\mathbf{w}_1, \ldots, \mathbf{w}_m)^T = (\boldsymbol{v}_1, \ldots, \boldsymbol{v}_d)$.

Suppose $a_1, \ldots, a_{m_t}, b_1, \ldots, b_{m_t}$ are the $i^{th}$ components of $\mathbf{w}^1$ where $\mathbf{w}^1$ shares common coordinates with $\mathbf{w}^2$. $a_{m_t+1}, \ldots, a_{m_1}, b_{m_t+1}, \ldots, b_{m_2}$ are the rest components. We have $\sum_{j=1}^{m_t} a_j \leq 1$ and $\sum_{j=1}^{m_t} b_j \leq 1$. Suppose $x_1, \ldots, x_{m_t}$ are the $i^{th}$ components of $\mathbf{w}$ that share common coordinates with $\mathbf{w}^1$ and $\mathbf{w}^2$. From Theorem 6, we have $\sum_{j=1}^{m_t} x_j = 1$.

Thus,

$$\|\boldsymbol{v}_i - \boldsymbol{u}_i^1\|_2^2 + \|\boldsymbol{v}_i - \boldsymbol{u}_i^2\|_2^2$$

$$= \sum_{i=1}^{m_t} \left[ (x_i - a_i)^2 + (x_i - b_i)^2 \right] + \sum_{i=m_t+1}^{m_1} a_i^2 + \sum_{i=m_t+1}^{m_2} b_i^2$$

$$= 2 \sum_{i=1}^{m_t} \left[ x_i^2 - (a_i + b_i)x_i + \frac{a_i^2 + b_i^2}{2} \right] + \sum_{i=m_t+1}^{m_1} a_i^2 + \sum_{i=m_t+1}^{m_2} b_i^2$$

$$= 2 \sum_{i=1}^{m_t} \left[ (x_i - \frac{a_i + b_i}{2})^2 + \frac{(a_i - b_i)^2}{4} \right] + \sum_{i=m_t+1}^{m_1} a_i^2 + \sum_{i=m_t+1}^{m_2} b_i^2$$

$$\geq 2 \frac{\left(1 - \sum\limits_{i=1}^{m_t}(a_i + b_i)/2\right)^2}{m} + \frac{\sum\limits_{i=1}^{m_t}(a_i - b_i)^2}{2} + \sum_{i=m_t+1}^{m_1} a_i^2 + \sum_{i=m_t+1}^{m_2} b_i^2.$$

Thus, we have

$$\min_{\theta} \frac{\|\boldsymbol{v}_i - \boldsymbol{u}_i^1\|_2^2 + \|\boldsymbol{v}_i - \boldsymbol{u}_i^2\|_2^2}{\|\boldsymbol{u}_i^1 - \boldsymbol{u}_i^2\|_2^2}$$

$$= \frac{2 \dfrac{\left(1 - \sum\limits_{i=1}^{m_t}(a_i+b_i)/2\right)^2}{m} + \dfrac{\sum\limits_{i=1}^{m_t}(a_i-b_i)^2}{2} + \sum\limits_{i=m_t+1}^{m_1} a_i^2 + \sum\limits_{i=m_t+1}^{m_2} b_i^2}{\sum\limits_{i=1}^{m_t}(a_i - b_i)^2 + \sum\limits_{i=m_t+1}^{m_1} a_i^2 + \sum\limits_{i=m_t+1}^{m_2} b_i^2} \triangleq M.$$

Further, we have

$$M \leq \frac{2 \dfrac{\left(1 - \sum\limits_{i=1}^{m_t}(a_i+b_i)/2\right)^2}{m} + \dfrac{\sum\limits_{i=1}^{m_t}(a_i-b_i)^2}{2} + \dfrac{(1-\sum\limits_{i=1}^{m_t} a_i)^2}{m_1 - m_t} + \dfrac{(1-\sum\limits_{i=1}^{m_t} b_i)^2}{m_2 - m_t}}{\sum\limits_{i=1}^{m_t}(a_i - b_i)^2 + \dfrac{(1-\sum\limits_{i=1}^{m_t} a_i)^2}{m_1 - m_t} + \dfrac{(1-\sum\limits_{i=1}^{m_t} b_i)^2}{m_2 - m_t}}.$$

Denote $S_a \triangleq \sum\limits_{i=1}^{m_t} a_i, S_b \triangleq \sum\limits_{i=1}^{m_t} b_i$, then we have

$$M \leq \frac{2 \dfrac{(1-\frac{S_a+S_b}{2})^2}{m} + \dfrac{(S_a-S_b)^2}{2m_t} + \dfrac{(1-S_a)^2}{m_1-m_t} + \dfrac{(1-S_b)^2}{m_2-m_t}}{\dfrac{(S_a-S_b)^2}{m_t} + \dfrac{(1-S_a)^2}{m_1-m_t} + \dfrac{(1-S_b)^2}{m_2-m_t}}$$

$$= \frac{\dfrac{m_1}{m_t(m_1-m_t)}(1-S_a)^2 + \dfrac{m_2}{m_t(m_2-m_t)}(1-S_b)^2}{\dfrac{(S_a-S_b)^2}{m_t} + \dfrac{(1-S_a)^2}{m_1-m_t} + \dfrac{(1-S_b)^2}{m_2-m_t}}$$

$$\leq \frac{\sqrt{m_1(m_2-m_t)m_2(m_1-m_t)}}{\sqrt{m_1(m_2-m_t)m_2(m_1-m_t)} - (m_1-m_t)(m_2-m_t)}$$

$$= \frac{1}{1 - \sqrt{\dfrac{(m_1-m_t)(m_2-m_t)}{m_1 m_2}}}.$$

Now we completed the proof.

$\square$

**Lemma 20** (Hoeffding's inequality). *Let $X_1, \ldots, X_n$ be independent random variables. Assume that $X_i \in [m_i, M_i]$ for every $i$. Then, for any $t > 0$, we have*

$$\mathbb{P}\left\{ \sum_{i=1}^{n}(X_i - \mathbb{E}X_i) \geq t \right\} \leq e^{-\frac{2t^2}{\sum_{i=1}^{n}(M_i - m_i)^2}}.$$

We suppose $m_1 \geq m_2$, then we have $\frac{m_t}{m_1} = \frac{\sum_{j=1}^{m_1} \mathbb{1}\{\mathbf{w}_j^2 \in S_i\}}{m_1}$. Then, from Hoeffding's inequality (Lemma 20), we have for any $\epsilon > 0$, with probability $1 - e^{-2m_1\epsilon^2}$,

$$\frac{\sum_{j=1}^{m_1} \mathbb{1}\{\mathbf{w}_j^2 \in S_i\}}{m_1} - \frac{1}{m+1} \geq -\epsilon.$$

Then we have

$$\frac{1}{1 - \sqrt{\frac{(m_1-m_t)(m_2-m_t)}{m_1 m_2}}} \leq \frac{1}{1 - \frac{m_1-m_t}{m_1}} = \frac{1}{\frac{m_t}{m_1}} \leq \frac{1}{\frac{1}{M+1} - \epsilon}.$$

Now, we suppose $\theta_1$ and $\theta_2$ respectively have $m_{1i}$ and $m_{2i}$ neurons that belong to $S_i$, and from Lemma 19 we have with probability $1 - \sum_{i=1}^{M} e^{-2 \max\{m_{1i}, m_{2i}\}\epsilon^2}$,

$$\min_{\theta} \frac{\|\boldsymbol{v}_i - \boldsymbol{u}_i^1\|_2^2 + \|\boldsymbol{v}_i - \boldsymbol{u}_i^2\|_2^2}{\|\boldsymbol{u}_i^1 - \boldsymbol{u}_i^2\|_2^2} \leq \frac{1}{\frac{1}{M+1} - \epsilon}$$

for all $i \in \{1, \dots, M\}$. Hence,

$$\min_{\theta} \frac{\|\theta - \theta_1\|_2^2 + \|\theta - \theta_2\|_2^2}{\|\theta_1 - \theta_2\|_2^2} \leq \frac{1}{\frac{1}{M+1} - \epsilon}.$$

Furthermore, for any $\delta > 0$, from Hoeffding's inequality, we have with probability $1 - e^{-2m\delta^2}$,

$$\frac{m_{1i}}{m} - \frac{1}{M+1} \geq -\delta.$$

Hence, with probability $1 - Me^{-2m\delta^2}$,

$$\frac{\max\{m_{1i}, m_{2i}\}}{m} - \frac{1}{M+1} \geq -\delta$$

for all $i \in \{1, \dots, M\}$. Therefore, with probability $1 - Me^{-2m\delta^2} - Me^{-2(\frac{m}{M+1} - m\delta)\epsilon^2}$,

$$\min_{\theta} \frac{\|\theta - \theta_1\|_2^2 + \|\theta - \theta_2\|_2^2}{\|\theta_1 - \theta_2\|_2^2} \leq \frac{1}{\frac{1}{M+1} - \epsilon}.$$

We let $\delta = \epsilon$, and we obtain with probability $1 - Me^{-2m\epsilon^2} - Me^{-2(\frac{m}{M+1} - m\epsilon)\epsilon^2}$,

$$\min_{\theta} \frac{\|\theta - \theta_1\|_2^2 + \|\theta - \theta_2\|_2^2}{\|\theta_1 - \theta_2\|_2^2} \leq \frac{1}{\frac{1}{M+1} - \epsilon}.$$

Last, using Cauchy's inequality, we completed the proof. $\qquad\square$

## A.7  Proof of Theorem 14

**Lemma 21.** *Suppose $a_1, \dots, a_{m_t}$, $b_1, \dots, b_{m_t}$ are the $i^{th}$ components of $\mathbf{w}^1$ and $\mathbf{w}^2$ where $\mathbf{w}^1$ shares common coordinates with $\mathbf{w}^2$ or where the neuron of $\mathbf{w}^1$ or $\mathbf{w}^2$ belongs to $S_0$. $a_{m_t+1}, \dots, a_{m_1}$, $b_{m_t+1}, \dots, b_{m_2}$ are the rest components. Then we have*

$$\min_{\theta} \frac{\|\boldsymbol{v}_i - \boldsymbol{u}_i^1\|_2^2 + \|\boldsymbol{v}_i - \boldsymbol{u}_i^2\|_2^2}{\|\boldsymbol{u}_i^1 - \boldsymbol{u}_i^2\|_2^2} = \frac{2\frac{\left(1 - \sum_{i=1}^{m_t}(a_i+b_i)/2\right)^2}{m_t} + \frac{\sum_{i=1}^{m_t}(a_i-b_i)^2}{2} + \sum_{i=m_t+1}^{m_1} a_i^2 + \sum_{i=m_t+1}^{m_2} b_i^2}{\sum_{i=1}^{m_t}(a_i - b_i)^2 + \sum_{i=m_t+1}^{m_1} a_i^2 + \sum_{i=m_t+1}^{m_2} b_i^2}.$$

**Proof of Lemma 21.** We have $\sum_{j=1}^{m_t} a_j \leq 1$ and $\sum_{j=1}^{m_t} b_j \leq 1$. Suppose $x_1, \ldots, x_{m_t}$ are the $i^{th}$ components of $\mathbf{w}$ that share common coordinates with $\mathbf{w}^1$ and $\mathbf{w}^2$. From Theorem 6, we have $\sum_{j=1}^{m_t} x_j = 1$. Thus,

$$
\begin{aligned}
&\|\boldsymbol{v}_i - \boldsymbol{u}_i^1\|_2^2 + \|\boldsymbol{v}_i - \boldsymbol{u}_i^2\|_2^2 \\
=& \sum_{i=1}^{m_t}[(x_i - a_i)^2 + (x_i - b_i)^2] + \sum_{i=m_t+1}^{m_1} a_i^2 + \sum_{i=m_t+1}^{m_2} b_i^2 \\
=& 2\sum_{i=1}^{m_t}[x_i^2 - (a_i + b_i)x_i + \frac{a_i^2 + b_i^2}{2}] + \sum_{i=m_t+1}^{m_1} a_i^2 + \sum_{i=m_t+1}^{m_2} b_i^2 \\
=& 2\sum_{i=1}^{m_t}[(x_i - \frac{a_i + b_i}{2})^2 + \frac{(a_i - b_i)^2}{4}] + \sum_{i=m_t+1}^{m_1} a_i^2 + \sum_{i=m_t+1}^{m_2} b_i^2 \\
\geq& 2\frac{\left(1 - \sum_{i=1}^{m_t}(a_i + b_i)/2\right)^2}{m_t} + \frac{\sum_{i=1}^{m_t}(a_i - b_i)^2}{2} + \sum_{i=m_t+1}^{m_1} a_i^2 + \sum_{i=m_t+1}^{m_2} b_i^2
\end{aligned}
$$

Thus, we have

$$
\min_{\theta} \frac{\|\boldsymbol{v}_i - \boldsymbol{u}_i^1\|_2^2 + \|\boldsymbol{v}_i - \boldsymbol{u}_i^2\|_2^2}{\|\boldsymbol{u}_i^1 - \boldsymbol{u}_i^2\|_2^2} = \frac{2\frac{\left(1 - \sum_{i=1}^{m_t}(a_i + b_i)/2\right)^2}{m_t} + \frac{\sum_{i=1}^{m_t}(a_i - b_i)^2}{2} + \sum_{i=m_t+1}^{m_1} a_i^2 + \sum_{i=m_t+1}^{m_2} b_i^2}{\sum_{i=1}^{m_t}(a_i - b_i)^2 + \sum_{i=m_t+1}^{m_1} a_i^2 + \sum_{i=m_t+1}^{m_2} b_i^2}.
$$

$\square$

**Lemma 22.** *With probability $1 - 5e^{-2m_t \epsilon^2}$,*

$$
\min_{\theta} \frac{\|\boldsymbol{v}_i - \boldsymbol{u}_i^1\|_2^2 + \|\boldsymbol{v}_i - \boldsymbol{u}_i^2\|_2^2}{\|\boldsymbol{u}_i^1 - \boldsymbol{u}_i^2\|_2^2} \leq A.
$$

*Here,*

$$
A = \frac{\frac{2\left(\frac{1}{m_t} - \frac{1}{2m_1} + \frac{1}{2m_2} + \epsilon\right)^2}{m_t} + \frac{c - \epsilon}{2} + (\frac{1}{m_t} - \frac{1}{m_1} + \epsilon)^2 + (\frac{1}{m_t} - \frac{1}{m_2} + \epsilon)^2}{c - \epsilon + (\frac{1}{m_t} - \frac{1}{m_1} + \epsilon)^2 + (\frac{1}{m_t} - \frac{1}{m_2} + \epsilon)^2}.
$$

**Proof of Lemma 22.** From Lemma 21, we have

$$
\min_{\theta} \frac{\|\boldsymbol{v}_i - \boldsymbol{u}_i^1\|_2^2 + \|\boldsymbol{v}_i - \boldsymbol{u}_i^2\|_2^2}{\|\boldsymbol{u}_i^1 - \boldsymbol{u}_i^2\|_2^2} \leq \frac{2\frac{\left(1 - \sum_{i=1}^{m_t}(a_i + b_i)/2\right)^2}{m_t} + \frac{\sum_{i=1}^{m_t}(a_i - b_i)^2}{2} + (1 - \sum_{i=1}^{m_t} a_i)^2 + (1 - \sum_{i=1}^{m_t} b_i)^2}{\sum_{i=1}^{m_t}(a_i - b_i)^2 + (1 - \sum_{i=1}^{m_t} a_i)^2 + (1 - \sum_{i=1}^{m_t} b_i)^2}.
$$

Since $\theta_1$ and $\theta_2$ are independently drawn from $SP(\mathcal{M}, r)$, $(a_i - b_i)^2$ has a positive expectation $c > 0$. Then, from Hoeffding's inequality (Lemma 20), for any $\epsilon > 0$, we have with probability $1 - e^{-2m_t \epsilon^2}$,

$$
\sum_{i=1}^{m_t}(a_i - b_i)^2 - cm_t \geq -m_t \epsilon.
$$

Similarly, we have with probability $1 - 2e^{-2m_t \epsilon^2}$,

$$\left|\sum_{i=1}^{m_t} a_i - \frac{m_t}{m_1}\right| \le m_t\epsilon,$$

and with probability $1 - 2e^{-2m_t\epsilon^2}$,

$$\left|\sum_{i=1}^{m_t} b_i - \frac{m_t}{m_2}\right| \le m_t\epsilon.$$

Hence, we have with probability $1 - 5e^{-2m_t\epsilon^2}$,

$$\min_\theta \frac{\|\boldsymbol{v}_i - \boldsymbol{u}_i^1\|_2^2 + \|\boldsymbol{v}_i - \boldsymbol{u}_i^2\|_2^2}{\|\boldsymbol{u}_i^1 - \boldsymbol{u}_i^2\|_2^2}$$

$$= \frac{2\dfrac{\left(1 - \displaystyle\sum_{i=1}^{m_t}(a_i+b_i)/2\right)^2}{m_t} + \dfrac{\displaystyle\sum_{i=1}^{m_t}(a_i-b_i)^2}{2} + \displaystyle\sum_{i=m_t+1}^{m_1} a_i^2 + \displaystyle\sum_{i=m_t+1}^{m_2} b_i^2}{\displaystyle\sum_{i=1}^{m_t}(a_i-b_i)^2 + \displaystyle\sum_{i=m_t+1}^{m_1} a_i^2 + \displaystyle\sum_{i=m_t+1}^{m_2} b_i^2}$$

$$\le A.$$

$\square$

Now, go back to the original problem, we have

$$\frac{m_t}{m_1} \ge \frac{\displaystyle\sum_{j=1}^{m_1} \mathbb{1}\{\mathbf{w}_j^2 \in S_i\} + \displaystyle\sum_{j=1}^{m_1} \mathbb{1}\{\mathbf{w}_j^2 \in S_0\}}{m_1}.$$

Then, from Hoeffding's inequality (Lemma 20), we have for any $\epsilon > 0$, with probability $1 - 2e^{-2m_1\epsilon^2}$,

$$\left|\frac{\displaystyle\sum_{j=1}^{m_1} \mathbb{1}\{\mathbf{w}_j^2 \in S_i\} + \displaystyle\sum_{j=1}^{m_1} \mathbb{1}\{\mathbf{w}_j^2 \in S_0\}}{m_1} - \frac{1 + (M-1)r}{M}\right| \le \epsilon.$$

Similarly we have with probability $1 - 2e^{-2m_2\epsilon^2}$,

$$\left|\frac{\displaystyle\sum_{j=1}^{m_2} \mathbb{1}\{\mathbf{w}_j^1 \in S_i\} + \displaystyle\sum_{j=1}^{m_2} \mathbb{1}\{\mathbf{w}_j^1 \in S_0\}}{m_2} - \frac{1 + (M-1)r}{M}\right| \le \epsilon.$$

Also, with probability $1 - e^{-2m\epsilon^2}$,

$$\frac{m_1}{m} - \frac{1 + (M-1)r}{M} \ge -\epsilon,$$

and with probability $1 - e^{-2m\epsilon^2}$,

$$\frac{m_2}{m} - \frac{1 + (M-1)r}{M} \ge -\epsilon.$$

We denote $\left(\frac{\frac{(M-1)(1-r)}{M} + \epsilon}{(\frac{1+(M-1)r}{M} - \epsilon)^2 m} + \epsilon\right)^2 = q$, then in this case, we have

$$A \leq \frac{1}{2} + \frac{\frac{2q}{(\frac{1+(M-1)r}{M} - \epsilon)^2 m} + q}{c - \epsilon + 2q}.$$

Now, we suppose $\theta^1$ and $\theta^2$ respectively have $m_{1i}$ and $m_{2i}$ neurons that belong to $S_i$, and we have with probability $1 - \sum_{i=1}^{M} 11 e^{-2(\frac{1+(M-1)r}{M} - \epsilon)^2 m \epsilon^2}$,

$$\min_{\theta} \frac{\|\boldsymbol{v}_i - \boldsymbol{u}_i^1\|_2^2 + \|\boldsymbol{v}_i - \boldsymbol{u}_i^2\|_2^2}{\|\boldsymbol{u}_i^1 - \boldsymbol{u}_i^2\|_2^2} \leq \frac{1}{2} + \frac{\frac{2q}{(\frac{1+(M-1)r}{M} - \epsilon)^2 m} + q}{c - \epsilon + 2q}$$

for all $i \in \{1, \dots, M\}$.

Hence, with probability $1 - 11 M e^{-2(\frac{1+(M-1)r}{M} - \epsilon)^2 m \epsilon^2}$,

$$\min_{\theta} \frac{\|\theta - \theta^1\|_2^2 + \|\theta - \theta^2\|_2^2}{\|\theta^1 - \theta^2\|_2^2} \leq \frac{1}{2} + \frac{\frac{2q}{(\frac{1+(M-1)r}{M} - \epsilon)^2 m} + q}{c - \epsilon + 2q}.$$

At last, we used Cauchy's inequality and we completed the proof. $\qquad\square$

## B  Proofs in Section 4

Before delving into Theorem 16, we first consider the simplest case where $L = 2, d = 1$ for gaining some intuition of the proof technique.

**Proposition 23.** *Theorem 16 holds true for $L = 2, d = 1$.*

*Proof.* Let $\theta_i = (A_1^{(i)}, A_2^{(i)})$ for $i = 1, 2$ be the two global minima. We are aiming to find a $\theta^* = (A_1^{(3)}, A_2^{(3)})$ satisfying

- $\theta^*$ is a global minima: $A_1^{(3)} A_2^{(3)} = Q$;
- $\theta_1$ and $\theta_2$ are 2-PL connected through $\theta^*$: for any $t \in [0, 1]$ and $i = 1, 2$, we have

$$(t A_1^{(i)} + (1 - t) A_1^{(3)})(t A_2^{(i)} + (1 - t) A_2^{(3)}) = Q. \tag{16}$$

Noticing that $\theta_1$ and $\theta_2$ are global minima, we have $A_1^{(i)} A_2^{(i)} = Q$ for $i = 1, 2$. Combining this with equation 16 leads to

$$A_1^{(3)} A_2^{(i)} + A_1^{(i)} A_2^{(3)} = 2Q$$

The problem now is converted to finding $A_1^{(3)} \in \mathbb{R}^{1 \times m}, A_2^{(3)} \in \mathbb{R}^{m \times 1}$ satisfied the following properties:

$$\begin{cases} A_1^{(1)} A_2^{(3)} = 2Q - A_1^{(3)} A_2^{(1)} \\ A_1^{(2)} A_2^{(3)} = 2Q - A_1^{(3)} A_2^{(2)} \\ A_1^{(3)} A_2^{(3)} = Q \end{cases}$$

1. When $A_1^{(1)}, A_1^{(2)}$ are linearly independent, we choose $A_1^{(3)}$ linearly independent of both $A_1^{(1)}$ and $A_1^{(2)}$. Then the problem above turns into solving a set of linear equations for $A_2^{(3)}$. Since $m \geq 3$, we can find a proper solution and finish the proof.

2. Suppose $k A_1^{(1)} = A_1^{(2)}$, we consider the first two equations, $\theta^* = (A_1^{(3)}, A_2^{(3)})$ exists only if $A_1^{(3)}(k A_2^{(1)} - A_2^{(2)}) = (2k - 2)C$, otherwise the first two equations will draw a contradiction by multiplying $k$ times in both sides of the first equation. Noticing that if $k A_2^{(1)} - A_2^{(2)} = 0$, we have $Q = A_1^{(2)} A_2^{(2)} = k^2 A_1^{(1)} A_2^{(1)} = k^2 Q$, then $k = 1$ or $-1$. We note that we have assumed that $k \neq -1$, while for $k = 1$, $(A_1^{(1)}, A_1^{(2)}) = (A_1^{(2)}, A_2^{(2)})$ is a trivial case. Thus, with the condition $k A_2^{(1)} - A_2^{(2)} \neq 0$, we can find a solution $A_1^{(3)}$ for $A_1^{(3)}(k A_2^{(1)} - A_2^{(2)}) = (2k - 2)C$ that is linearly independent of $A_1^{(1)}, A_2^{(1)}$ (we can find $m - 1 \geq 2$ linearly independent solutions in fact) first, then get a proper $A_2^{(3)}$ and finish the whole proof.

□

*Remark* 24. This will not hold for the case that $(A_1^{(1)}, A_2^{(1)}) = (-A_1^{(2)}, -A_2^{(2)})$, since $k = -1, (kA_1^{(2)} - A_2^{(2)}) = 0$ will directly draw $0 = -4A_1^{(2)} A_2^{(2)}$, which will be a contradiction when $A_1^{(2)} A_2^{(2)} \neq 0$. We outline a simple counter-example here: $A_1^{(1)} = (1, 0, 0, 0), A_2^{(1)} = (1, 0, 0, 0)^T, A_1^{(2)} = (-1, 0, 0, 0), A_2^{(2)} = (-1, 0, 0, 0)^T, Q = 1$. In particular, denote the center $A_1' = (\alpha_1, \ldots, \alpha_4), A_2' = (\beta_1, \ldots, \beta_4)$, the necessary conditions can be converted into $\alpha_1 + \beta_1 = 2, \alpha_1 + \beta_1 = -2$, $\sum_{i=1}^4 \alpha_i \beta_i = 2$, which draw a contradiction.

**Proof of Theorem 16**

**The case of $2$-PL connectivity.** With the toy structure in Proposition 23, we can similarly consider the representation structure in Theorem 16 with the case $d > 1$.

*Proof.* If we consider each column of $A_2$ separately, then the problem turns out to be the case in Proposition 23 by considering each column separately. Once $A_1^{(1)}, A_1^{(2)}$ are independent, we consider $A_1^{(3)}$ linearly independent of $A_1^{(1)}, A_1^{(2)}$ and find a solution for each column in $A_2^{(3)}$ independently, then the problem is solved directly by applying the case with $d = 1$ separately.

Now we consider the case that $kA_1^{(1)} = A_1^{(2)}$ for some $k \in \mathbb{R}$, in this case by considering conditions

$$\begin{cases} A_1^{(1)} A_2^{(3)} = 2Q - A_1^{(3)} A_2^{(1)} \\ A_1^{(2)} A_2^{(3)} = 2Q - A_1^{(3)} A_2^{(2)} \end{cases}$$

we need $A_1^{(3)}(kA_2^{(1)} - A_2^{(2)}) = (2k - 2)Q$. We view it as solving linear equations $A_1^{(3)}(kA_2^{(1)} - A_2^{(2)}) = (2k - 2)Q$ for $A_1^{(3)}$. Denote $Q = (Q_1, \ldots, Q_n)$. If $\mathbf{rank}(kA_2^{(1)} - A_2^{(2)}) = d < m$, it would be easy to find a solution $A_1^{(3)}$ that is additionally independent to $A_1^{(1)}$. Otherwise, if $\mathbf{rank}(kA_2^{(1)} - A_2^{(2)}) < d$, without loss of generality, we consider a case of column-wise linear dependency. Assume that the first two columns of $A_2^{(1)}, A_2^{(2)}$ are $(\alpha_1, \alpha_2), (\beta_1, \beta_2)$, respectively. If $k\alpha_1 - \beta_1 = t(k\alpha_2 - \beta_2)$, we naturally have

$$k^2 Q_1 - Q_1 = A_1^{(2)}(k\alpha_1 - \beta_1) = tA_1^{(2)}(k\alpha_2 - \beta_2) = tk^2 Q_2 - tQ_2.$$

We have assumed that $k \neq -1$. If $k = 1$, the linear connectivity always holds naturally. On the other hand, if $k^2 \neq 1$, we will obtain $Q_1 = tQ_2$, which will never affect the linear equations in $A_1^{(3)}(kA_2^{(1)} - A_2^{(2)}) = (2k - 2)Q$.

Thus, we can find a solution $A_1^{(3)}$ that satisfies $A_1^{(3)}(kA_2^{(1)} - A_2^{(2)}) = (2k - 2)Q$ for $A_1^{(3)}$ and independent to $A_1^{(1)}$. With this, we can then derive a proper $A_2^{(3)}$ by considering each column separately as in Proposition 23. Then, we finish our proof of Theorem 16. □

**The case of $3$-PL connectivity.** For two minima

$$\theta_1 = (A_L^{(1)}, A_{L-1}^{(1)}, \ldots, A_1^{(1)}), \ \theta_2 = (A_L^{(2)}, A_{L-1}^{(2)}, \ldots, A_1^{(2)})$$

that belong to the zero-measure set in the proof of 2-PL connectivity, we consider $\theta_3 = (A_L', A_{L-1}^{(1)}, \ldots, A_1^{(1)})$, where $A_L'$ satisfies that $A_L' A_{L-1}^{(1)} \ldots A_1^{(1)} = Q$. Then it is natural that $\theta_1 \leftrightarrow \theta_3$.

We consider the following linear relationship: $A_L' A_{L-1}^{(1)} \ldots A_t^{(1)}$ and $A_L^{(2)} A_{L-1}^{(2)} \ldots A_t^{(2)}$ are linearly independent for $t = L, \ldots, 2$, which would yield 2-PL connectivity of $\theta_2$ and $\theta_3$ following by the discussion above. To satisfy the $L - 1$ conditions along with $A_L' A_{L-1}^{(1)} \ldots A_1^{(1)} = Q$, we consider seeking for $F_L, \ldots, F_2$ such that $F_j$ is independent of $A_{L-1}^{(2)} \ldots A_j^{(2)}$ for $j = 2, \ldots, L$. Then we consider $A_L' A_{L-1}^{(1)} \ldots A_j^{(1)} = F_j$ for $j = 2, \ldots, L$ and $A_L' A_{L-1}^{(1)} \ldots A_1^{(1)} = Q$ as $L$ linear equations on the $m_1$ parameters in $A_L'$. We have assumed that $m > 2L - 1$, thus if $E_t = A_{L-1}^{(1)} \ldots A_t^{(1)}$, $t = L, \ldots, 1$ ($E_L = \mathbf{1}_m$) are linearly independent, the linear system would have a proper solution.

On the other hand, as we need to select $F_L, \ldots, F_2$ to make sure the linear equations have a solution, we consider solving the dependency by the following adjustments. If some of the $E_j$'s are linearly dependent, for instance, $E_k$ can be represented by the linear combination of another group of $E_j$'s, then our $F_k$ need to be automatically determined by the corresponding $F_j$'s to ensure the existence of solutions. Since $F_k$ need to be independent of $A_{L-1}^{(2)} \ldots A_k^{(2)}$, here we (might) lose a degree of freedom when selecting $F_j$'s, while the fact that we do not need to consider $F_k$ anymore reduces an equation from the $L$ conditions required. Thus, in the process of reducing our restrictions to be linearly independent, our degree of freedom would always be greater than the number of restrictions since $m > 2L - 1$ at

the beginning. Thus, with the (eventually) linear-independent coefficients (with the number of restrictions less than parameters), we will have a proper solution of $A'_L$ that meets the above conditions, which would yield $\theta_2 \leftrightarrow \theta^*, \theta^* \leftrightarrow \theta_3$ for some $\theta^*$ from the first statement. Thus we will have $\theta_1 \leftrightarrow \theta_3, \theta_3 \leftrightarrow \theta_*, \theta_* \leftrightarrow \theta_2$. □

## B.1 Proof of Theorem 18

We first propose a lemma about linear independence which we will use in our later proof.

**Lemma 25.** *For linearly independent vectors* $C_1, \ldots, C_p \in \mathbb{R}^{1 \times r_1}$, *linearly independent vectors* $D_1, \ldots, D_q \in \mathbb{R}^{1 \times r_2}$, *and vectors* $E_1, \ldots, E_p \in \mathbb{R}^{1 \times r_2}$, *if* $r_2 > p + q$ *and* $r_1 > q$, *there exist a matrix* $K \in \mathbb{R}^{r_1 \times r_2}$ *such that the* $p + q$ *vectors* $C_i K + E_i (i = 1, 2, \ldots, p)$, $D_i (i = 1, 2, \ldots, q)$ *are linearly independent.*

*Proof.* Since $r_2 > p + q$, we can find another $p$ vectors $F_1, \ldots, F_p$ such that $D_1, \ldots, D_q, F_1, \ldots, F_p$ are linearly independent.

Then, we consider the $p$ equations $C_i K + E_i = F_i (i = 1, 2, \ldots, p)$. Since $C_1, \ldots, C_p \in \mathbb{R}^{1 \times r_1}$ are independent and $r_1 > p$, we can find a solution for $K$ by considering each column of $K$ separately.

Thus, with $C_i K + E_i = F_i (i = 1, 2, \ldots, p)$ and the linear independence of $D_1, \ldots, D_q, F_1, \ldots, F_p$, we finish our proof of the lemma. □

Now we begin with our proof of 18.

*Proof.* Following the explicit description in equation 5, it is natural to consider whether the explicit representation of linear layers achieves $A_L^* \ldots A_2^* A_1^*$. As a set out of only zero-measure points in the minima manifold, we consider an assumption below:

**Assumption 26.** For all $k = 1, 2, \ldots, n - 1$, the $r$ vectors $A_L^{(i)} \ldots A_2^{(i)} A_1^{(i)} \in \mathbb{R}^{1 \times m} (i = 1, 2, \ldots, r)$ are linearly independent.

To begin, for each $i \in [1, r], q = 1, 2, \ldots, L$, we define $\sigma_{i,k,q}$ to be the sum of all $C_q^k$ elements in

$$\left\{ B_L B_{L-1} \ldots B_{L-q+1} \middle| \text{for } j \in \{1, \ldots, q\}, \text{only } k \text{ of } B_j \text{ to be } A_j^i, \text{ while the remaining } q - k \ B_j \text{ to be } A_j^* \right\}, \quad (17)$$

With this notation, our desirable connectivity property can be written in terms that

$$(tA_L^{(i)} + (1-t)A_L^*)(tA_{L-1}^{(i)} + (1-t)A_{L-1}^*) \ldots (tA_1^{(i)} + (1-t)A_1^*) = \sum_{j=0}^{L} t^j (1-t)^{L-j} \sigma_{i,j,L}$$

for all $i = 1, 2, \ldots, r$ and $j = 1, 2, \ldots, L$. Since this should be right for any $t \in [0, 1]$ in the context of star-shaped connectivity, it is natural to find $A_1^*, \ldots, A_L^*$ such that $\sigma_{i,k,L} = C_L^k Q$ for all $i$ and $k$, which are the necessary condition followed by considering different order terms of $t$. On the other hand, if this holds, we will directly derive that each point in the star-shaped manifold is an exact minimum, i.e., for any $t \in [0, 1]$,

$$(tA_L^{(i)} + (1-t)A_L^*)(tA_{L-1}^{(i)} + (1-t)A_{L-1}^*) \ldots (tA_1^{(i)} + (1-t)A_1^*) = \sum_{j=0}^{L} C_L^j t^j (1-t)^{L-j} Q = Q.$$

Now we start to construct $A_1^*, \ldots, A_L^*$ inductively. Firstly, consider the case in Assumption 26 when $k = 1$. Since $m > r + 1$, we can find $A_L^*$ by Lemma 25 such that $\mathcal{C}_1 = \{A_L^*, A_L^{(1)}, \ldots, A_L^{(r)}\}$ are linearly independent.

Then, we consider Assumption 26 with $k = 2$. We notice that $A_L^*, A_L^{(1)}, \ldots, A_L^{(r)}$ are linearly independent and $m > 2r + 1, m > r$, then following Lemma 25, we can find $A_{L-1}^*$ such that the $2r + 1$ vectors in

$$\mathcal{C}_{L-1} = \left\{ A_L^{(i)} A_{L-1}^* + A_L^* A_{L-1}^{(i)} (i = 1, 2, \ldots, r), A_L^{(i)} A_{L-1}^{(i)} (i = 1, 2, \ldots, r), A_L^* A_{L-1}^* \right\}$$

are linearly independent.

We repeat the process using Assumption 2,3 and Lemma 25 to accumulate more layers while keeping with their independent property. In particular, with $wr + 1(1 < w < L - 1)$ linearly independent vectors in

$$\mathcal{C}_w = \{\sigma_{i,k,w-1} A_{L-w+1}^* + \sigma_{i,k-1,w-1} A_{L-w+1}^i (k = 1, \ldots, w - 1; i = 1, \ldots, r),$$

$$A_L^* A_{L-1}^* \dots A_{L-w+1}^*, A_L^{(i)} A_{L-1}^{(i)} \dots A_{L-w+1}^{(i)} (i = 1, 2, \dots, r)\},$$

we consider Lemma 25 to obtain $K$ as $A_{L-w}^*$, with $D_i$ being independent vectors $A_L^{(i)} \dots A_1^{(i)}$, $i = 1, 2, \dots, r$ being ensured by Assumption 26. By doing so, we derive a new group of independent vectors

$$\mathcal{C}_{w+1} = \{\sigma_{i,k,w} A_{L-w}^* + \sigma_{i,k-1,w} A_{L-w}^i (k = 1, \dots, w; i = 1, \dots, r), A_L^* A_{L-1}^* \dots A_{L-w}^*, A_L^{(i)} A_{L-1}^{(i)} \dots A_{L-w}^i (i = 1, 2, \dots, r)\}.$$

Thus, by induction, we can sequentially get $A_L^*, \dots, A_2^*$ such that all $(L-1)r + 1$ vectors

$$\mathcal{C}_{L-1} = \{\sigma_{i,k,L-2} A_2^* + \sigma_{i,k-1,L-2} A_2^i (k = 1, \dots, L-2; i = 1, \dots, r),$$
$$A_L^* A_{L-1}^* \dots A_2^*, A_L^{(i)} A_{L-1}^{(i)} \dots A_2^i (i = 1, 2, \dots, r)\}$$

are linearly independent.

We illustrate that keeping such linearly independent properties in induction is of essential importance in this proof, which shares the very essence in Theorem 16, providing a sufficient condition for us to obtain the proper solution of the linear equations.

Finally, recall that we aim to make sure $\sigma_{i,k,L} = C_L^k Q$ for all $i = 1, 2, \dots, r$ and $k = 0, 1, \dots, L$. Consider that the case for $k = L$ is naturally true, and all $r$ equations when $k = 0$ are the same, we can therefore view that as $(L-1)r + 1$ linear equations of variable $A_1^*$ since $A_L^*, \dots, A_2^*$ have been fixed. Since all the coefficients of $A_1^*$ in the $(L-1)r + 1$ equations are actually the $(L-1)r + 1$ vectors in $\mathcal{C}_{L-1}$, which are linearly independent, we can therefore find a proper solution for $A_1^*$ by considering each column in $A_1^*$ separately. The condition $m > (L-1)r + 1$ in Assumption 2 makes sure of the existence of the solution.

Thus, we finish the construction of $A_L^*, \dots, A_1^*$, which satisfies the required property, and finish the proof.

$\square$

