# OpenReview forum: "Exploring Neural Network Landscapes: Star-Shaped and Geodesic Connectivity"
_TMLR — Withdrawn by Authors_

### Review · Reviewer_TK2h · 2024-06-10

**Summary Of Contributions:**

This paper theoretically and empirically analyzes mode connectivity, the ability to interpolate between different neural network minima and achieve low-loss along the entire path. On the theoretical side, it proves under some technical assumptions in simple settings that one should expect mode connectivity for piecewise paths, and as a result that the loss landscape is star convex, where any group of minima can be connected through a central minimum. On the empirical side it confirms the star convex behavior in small-scale image classification.

**Audience:**

Yes

**Claims And Evidence:**

Yes

**Requested Changes:**

Please see Weaknesses. In particular the most important ones are intuition behind the theory, clarification of the theoretical setting and expanded discussion in the conclusion. Also seem minor issues below.

Minor Issues:
- "Here, B_r, B_t \in \mathbb{N} denote the batch sizes..." this is somewhat confusing as typically batch size is used to denote the number of samples (and it is later used like this), it would be helpful to specify that B_r corresponds to the number of minima, while B_t corresponds to the number of interpolated points.
- some typos
- A number of places mention "linear mode connectivity" when really "piecewise linear mode connectivity" or "k-linear mode connectivity" is what should be used (e.g. caption of Table 2)
- The use of the word "Barrier" in Table 1 and Table 2 is not consistent with prior literature on mode connectivity [1, 2]. Typically barrier means the difference between the maximum/minimum along the interpolation and the endpoints, while here it means just the maximum/minimum along the interpolation. I think it would be clearer to use "Maximum Loss" and "Minimum Accuracy"

**Strengths And Weaknesses:**

### Strengths:
- Empirically, star convexity is a new property, and it's interesting to see that the landscape is connected this way.
- There was exploration both on the theoretical and empirical side, which is appreciated (the experiments are not left to the reader).

### Weaknesses:
- A major weakness of this paper is that often the theoretical results are presented without an explanation as to why they might be true, or a proof sketch. Such presentation makes it difficult to disentangle whether the results have the potential to apply more broadly or to more practical networks.
- I am somewhat unaware of the theoretical literature, but on a quick review it seems that the definition of 2-layer neural network is not always consistent [1, 3, 6, 7] vs. [5]. The one this paper uses only has 1 matrix unlike the common definition W_1 \sigma(W_2x) [1, 3, 6, 7]. I believe Thm. 6 (and thus many of the theoretical results) depends on the fact that the network does not have a 2nd matrix as otherwise there would be a counterexample by flipping signs, which I believe makes the setting quite restrictive and less likely to apply in practice.
- As I understand the major theoretical argument here depends on sufficiently large width, and thus excess capacity that one can interpolate in, which seems very similar in spirit to the "dropout stability" assumption, and proof argument of Kuditipudi et al. [4]. As that work is done for more complicated and deeper networks, it's not clear to me the major difference. In particular "dropout stability" seems to imply the student teacher setting in a much less restrictive case (the teacher can be half the width), so it would be helpful to have clarification around the results.
- In Thm. 14, it is assumed that there is a uniform sparsity on minima, but these minima aren't ever shown to be reachable by optimization, nor is there any empirical confirmation that such uniform sparsity is an expected pattern (Fig. 3 just shows sparsity is possible), so I'm not sure that this "demonstrates that the bias towards sparse solutions can explain the shrinkage of NGD to 1."
- I feel the phrase "We independently train the model to find 3 minima" is missing many details. For example: how were networks initialized? Were hyperparameters the same except random seed and initialization? Was anything shared between runs? These precise details seem particularly important here given the linear mode connectivity phenomenon relies on sharing parameters and differing seeds [2].
- This is somewhat minor, but I feel strange about the claim of star convexity, as that might imply some sort of central minimum that all minima must pass through, yet the way to find this central minimum is post-hoc and the experiments are only performed for 3 checkpoints. I realize that these experiments may be somewhat expensive for larger nets, but it would be nice to plot some metric of star convexity vs. the number of minima and see if or where the predictions break down. As an example see [9] on generalizing LMC to multiple dimensions.
- The method for finding the central minimum is quite close to that employed in [8]. That work shows entire subspace connectivity, though without theoretical grounding, which may provide for interesting discussion in the current text.
- I think it would be nice to make comment about what the fact that NGD going to 1 implies. If LMC does not hold, but there is a very-close-to-linear path on which one can interpolate, that means the nonlinearity is pushed into some few dimensions. I think the discussion after Theorem 14 could be expanded in the conclusion to give a higher level picture as to the speculation why this might be possible.

References:
1. Arora et al. Fine-Grained Analysis of Optimization and Generalization for Overparameterized Two-Layer Neural Networks. 2019. https://arxiv.org/abs/1901.08584
2. Frankle et al. Linear Mode Connectivity and the Lottery Ticket Hypothesis. 2019. https://arxiv.org/abs/1912.05671
3. Ghorbani et al. Limitations of Lazy Training of Two-layers Neural Networks. 2019. https://arxiv.org/abs/1906.08899
4. Kuditipudi et al. Explaining Landscape Connectivity of Low-cost Solutions for Multilayer Nets. 2019. https://arxiv.org/abs/1906.06247
5. Li and Yuan. Convergence Analysis of Two-layer Neural Networks with ReLU Activation. 2017. https://arxiv.org/abs/1705.09886
6. Mei et al. A mean field view of the landscape of two-layer neural networks. 2018. https://www.pnas.org/doi/abs/10.1073/pnas.1806579115
7. Nguyen and Woodrow. Improving the Learning Speed of 2-Layer Neural Networks by Choosing Initial Values of the Adaptive Weights. 1990. https://ieeexplore.ieee.org/abstract/document/5726777
8. Wortsman et al. Learning Neural Network Subspaces. 2021. https://arxiv.org/abs/2102.10472
9. Yunis et al. On Convexity and Linear Mode Connectivity in Neural Networks. 2022. https://openreview.net/forum?id=TZQ3PKL3fPr

---

### Review · Reviewer_ACMH · 2024-06-18

**Summary Of Contributions:**

The paper explores the phenomenon of star-shaped connectivity in overparameterized neural networks. It demonstrates that for any finite set of global minima, there exists a central minimum that is linearly connected to all the models in the set. The main contributions of the paper are as follows:
1) The paper provides a theoretical foundation for star-shaped connectivity in two-layer ReLU networks and deep linear networks.
2) It introduces the concept of geodesic connectivity, showing that the geodesic distance between two models on the minima manifold is close to the Euclidean distance between the points. Furthermore, this ratio approaches 1 as the network width increases.
3) The paper presents an SGD-based algorithm for finding the central network that is linearly connected to a predefined set of minima.
4) The paper validates star-shaped and geodesic connectivity using practical experiments on the MNIST and CIFAR-10 datasets.

**Audience:**

Yes

**Broader Impact Concerns:**

To the best of my knowledge, this paper does not have any potential negative impacts. Furthermore, studies on the loss landscape, such as the one presented here, contribute significantly to our understanding of neural networks. This understanding is crucial for developing robust and secure recognition systems.

**Claims And Evidence:**

Yes

**Requested Changes:**

**Critical concerns**

1) More experimental verification of star-shaped and geodesic connectivity on larger datasets (with higher resolution and a greater number of samples) and different neural network architectures (e.g., transformers and various convolutional architectures) is required. I recommend making this set of experiments more practical by using the appropriate optimizer and augmentations for each network.

2) I suggest comparing test metrics of minima obtained with SGD and the center minima, as well as the barriers between them, similar to the analysis presented in Figure 5.

3) The proof for Theorem 16 includes only the case of $L=2$ but larger depth values are not considered.

**Strengthening adjustments**
1) The behavior of well-known objects in high-dimensional spaces is often counterintuitive. We see that for higher dimensions (i.e., wider networks) NGD approaches 1. The authors claim that this is the property of the loss landscape, which becomes “more convex”. But what if this is the property of high-dimensional space itself (i.e., similarly to the dimensionality curse)? I would recommend the authors to elaborate on this.

2) Following the previous comment, I think it makes sense to plot 2D visualization of the loss landscape spanned over 2 SGD minima and the center model (similarly to Figure 1 in https://arxiv.org/pdf/1802.10026) for different values of neural network width. This visualization might support the argument that the landscape becomes “more convex”.

3) The theorems for two-layer ReLU networks hold when the width of the network $m$ is (significantly) larger than the width of the teacher network $M$. Is it possible to say anything about minima and their mutual arrangement in the weight space for the case of underparameterized setup $m \le M$?

4) Making the proofs more detailed and describing each particular step would also be beneficial.

**Minor notes and questions**
1) It seems that it should be $\alpha > 0$ in the definition of $S_j$ in Theorem 6 (at least it is shown that $w_{ij} \ge 0$ (equation 12), so negative values of $\alpha$ are impossible.
2) Should it be $M+1$ types of neurons instead of $m+1$ (first bullet, top of page 5)?
3) Should it be $\theta$ instead of $\theta_t$ in equation 7?

**Strengths And Weaknesses:**

**Strengths**

1) The proposed phenomenon of star-shaped connectivity is novel and broadens the understanding of loss landscaped of deep neural networks

2) The theoretical aspects of the paper are highly relevant. The assumptions are mostly adequate, and the proofs appear to be correct and comprehensive, with minor exceptions noted in the Requested Changes section.

3) Although the theoretical setup with two-layer ReLU networks is rather specific, it provides convincing insights. The proof offers an intuitive understanding of why star-shaped connectivity arises in overparameterized networks: the structure of input data and target variables determines the types of neurons (the set $S$) present in the minima. If the network is sufficiently wide, there is a high probability that some neuron types in different minima are implemented with the same neurons, allowing the networks to be connected with a two-segment path.

4) The practical section of the paper is clearly written and includes a detailed description of the experimental setup, ensuring the study can be reproduced.

**Weaknesses**
1) The experimental section of the paper is relatively limited. Additional experiments are necessary to robustly confirm the star-shaped connectivity hypothesis. These should include tests on larger datasets and various neural network architectures. Additionally, the optimization setup used in the paper (Adam without augmentations) differs from the conventional setup for convolutional networks (SGD with momentum and augmentations).

2) Generalization is a critical factor when comparing different optima obtained with SGD. However, the experimental section of the paper reports the loss and accuracy barriers only for the training set, neglecting the evaluation on the test set.

3) Some theoretical proofs are not detailed enough and thus are difficult to follow. For example, it makes sense to write $\theta^*$ in a closed form in the proof of Theorem 8.

---

### Review · Reviewer_Fa4w · 2024-06-19

**Summary Of Contributions:**

The paper investigates the previously observed "mode connectivity" properties of neural networks. Specifically, it provides theoretical justifications that, for linear networks and two-layer ReLU networks in a teacher-student setup, the connecting path can be a simple two-piece linear path (which is only slightly longer than their direct connection). Furthermore, it discovers that there exists a "center" point, on the minima manifold, that connects to all minima via linear paths. These theoretical results are supported by empirical evidence on MNIST and CIFAR-10.

**Audience:**

Yes

**Broader Impact Concerns:**

As this paper describes foundational research without any application, I don't believe a dedicated discussion of the broader ethical implications is necessary.

**Claims And Evidence:**

Yes

**Requested Changes:**

General comments:

- I find the statement that an NGD close to 1 meaning that the landscape "should be nearly convex in a certain sense" to be quite vague. It only means that it is nearly convex in this particular direction and only near the studied minima, no? You also claim that "If the NGD keeps close to 1 for any pair of minima, then the landscape should be somehow as benign as a convex one. Otherwise, the landscape must be highly non-convex", then in Section 5.2 "It is demonstrated that minima [...] can be connected via paths whose NGDs are nearly 1. This is consistent with our theoretical findings in toy models.". Doesn't this contradict the very first sentence of the Introduction "It is well-known that neural networks are highly non-convex [...]"? If so, I think this contradiction deserves more attention.
- In general, I am not sure whether the term "minimum" is always precise. At least for the empirical results, it might be more appropriate to talk about "iterates/solutions found by SGD" or something similar.
- I find the term "practical models" (in the list of contributions) to be a bit of an overstatement for the models trained on MNIST and CIFAR-10.

Potential fixes:

- In the caption of Figure 1, right (and Figure 5), isn't the mathematical description of the orange line incorrect? Shouldn't it go from $\theta_1$ to $\theta_2$, i.e. $t \theta_1 + (1-t) \theta_2$, instead of linearly interpolating between the center and $\theta_i$ (so both minima?)? The same is in Figure 5.
- In Section 1.1 under Nonlocal structures, you write "...showed a star-shaped structure in the regime of the spherical negative perceptron. We instead [...] showing that [a] star-shaped structure also exists for neural networks". But isn't a perceptron also a neural network?
- At the top of page 5, should it be a capital $M$ instead? I.e. "There are at most $M+1$ types of student neurons$ instead of $m+1$?

Questions I had, where I would be interested in hearing the author's reply:

- How does the fact that NGD is close to 1 influence the star-shaped connectivity? If I understand correctly, this means that while the direct way between two minima has a significant loss barrier, to go around it only requires a slight detour. So does that mean that it is not a very "pointy" star shape?
- What does "typical" mean in the abstract in "For a finite number of typical minima,..."?
- Is this "center-point" of the star in any way special, e.g. having better generalization or anything like this?
- Did you investigate how many such center-points exist? It seems to me that there could be quite a lot of points with this property. With the penalization function in Eq. (6) you are probably biasing the optimizer to find a specific one. Can you elaborate a bit on this?
- Is Figure 2 (right) an empirical result? I think it is "just" an illustrative plot, but perhaps showing actual results (e.g. Figure 5) would be more informative and concise.
- To me, the theoretical assumptions sound quite unrealistic and very specific. For e.g. the two-layer ReLU network, only uses a single activation (no softmax output or similar), and the second "layer" is a simple sum without any weights, no? So it is a very special case of a two-layer ReLU network. Furthermore, I am a bit surprised by the student-teacher setting. First of all, the teacher's parameters are assumed to be orthonormal but more surprisingly for me, you always require that the student has a larger number of neurons than the teacher. For me, this is a surprising setting, that the student has a higher capacity than the teacher. You also assume a squared loss and a single output, e.g. regression, no? Could you elaborate perhaps on the theoretical assumptions, how common they are in the literature, how realistic they are to be fulfilled in practice, how limiting they are, etc.? If the assumptions are indeed quite specific, perhaps it would be more appropriate then to write something like "These results are provably valid for linear networks and specific two-layer ReLU networks under a special teacher-student setup". I think that this limitation (if it is true) should be mentioned honestly and transparently in the abstract and the main text.
- Below Assumption 15 you say "The above assumption is quite mild" but doesn't this assumption mean that the input data is linearly connected to the output data (and as you mentioned, consequently you can simply fit it with your linear network)? To me, this sounds like a very strong assumption. Could you elaborate on why you think it is a mild assumption?

Minor suggestions to improve presentation or readability:

- Citations could be cleaned up. E.g. by using `\citet` (or `\textcite`) and `\citep` (or `\parencite`). E.g. in the third paragraph of the Introduction: "... was first observed by Freeman and Bruna in Freeman & Bruna (2017)" could be replaced with "...was first observed by Freeman & Bruna (2017)".
- In the "Geodesic connectivity" contribution, I think the second sentence might be grammatically incorrect. Perhaps it could be "...the ratio between them monotonically decreases towards 1 as the network width increases".
- Overall, the plots could be improved a bit. They tend to have lots of whitespace while having relatively small font sizes. Furthermore, the font size is quite inconsistent (e.g. comparing Fig 3 and 4). In general, the consistency across plots could be improved:
  - In Fig 3 left, the width is given as 2^m, while in Fig 4 it is described as log(width), which is a bit inconsistent.
  - Figures sometimes have titles, sometimes not. For example in Figure 4, the title is in my opinion redundant (with the legend), while Fig 1 left has a title but Fig 1 right does not.
- I always think that a "two-layer" neural network is used ambiguously in our community. Does this mean input -> hidden layer -> output, or input -> hidden layer 1 -> hidden layer 2 -> output? If I understood it correctly from the remaining context, in your case it is the former, but perhaps this could be highlighted somewhere.
- The paragraph between Theorem 10 and 11 might be grammatically wrong.
- Lemma 17 might be missing square brackets for the expectation at the very end (to indicate whether the square is inside or outside of the expectation).
- Nit: When typesetting "vs.", e.g. in caption 4, you should use "vs.~" in LaTeX, to avoid the larger space after the period.
- The notation is sometimes a bit inconsistent. E.g. In Section 5, you once use "lr" for the learning rate and once "$\eta$". Same with the center, which is sometimes $\theta^\ast$ (e.g. in the caption of Figure 5 or Definition 3), then just $\theta$ (e.g. in the center-finding algorithm in Section 5, where the minima are denoted with $\theta_i^\ast$ instead), and then also with $\theta_{c^\ast}$ in 5.1. This is a bit confusing.
- I believe "batchsize" should be "batch size" instead.

I did not check the proofs in the appendix for technical correctness as this falls outside of my area of expertise.

**Strengths And Weaknesses:**

### Strength

- The paper studies a very interesting and relevant topic. It clearly fulfills the scope of TMLR of being an "experimental and/or theoretical study yielding new insight into the design and behavior of learning in intelligent systems".
- It provides both theoretical and empirical results that strengthen each other.
- It extends the observed phenomenon of mode connectivity with interesting properties, such as the simple two-piece linear path connection or the star-shaped connectivity.
- Overall, it is easy to follow the paper's story and reasoning.

### Weaknesses

- The tested architectures are tiny. E.g. Figure 3 right uses an input dimension of 4 and 4 teacher neurons. The empirical results later, are also only for MNIST and CIFAR-10.
- It is unclear how robust the empirical results are. It would be great to have error bars, e.g. for Figure 4 (which only shows the average results of 5 experiments). Figure 5 is probably showing the results from just two SGD runs. Figure 5 visualizes star-shaped connectivity from just three minima (although Table 1 shows that overall 5 minima were trained). Overall, it would be helpful to repeat those experiments with more runs/minima to estimate how robust the results are, e.g. by reporting mean and variances.
- Some of the claims are phrased a bit too strongly for my taste (see more detailed comments below). E.g. as far as I can tell, the theoretical assumptions are quite strong, while e.g. the conclusion "We provide theoretical analysis on toy models such as two-layer ReLU networks and linear networks" makes it sound like the theoretical results apply very generally.

---

### Note · Authors · 2024-07-12

I have read and agree with the venue's withdrawal policy on behalf of myself and my co-authors.